



# Evolution of continental temperature seasonality from the Eocene greenhouse to the Oligocene icehouse – a model–data comparison

**Agathe Toumoulin**[1], **Delphine Tardif**[1,2], **Yannick Donnadieu**[1], **Alexis Licht**[1], **Jean-Baptiste Ladant**[3], **Lutz Kunzmann**[4], and **Guillaume Dupont-Nivet**[5,6]

[1]Aix Marseille Université, CNRS, IRD, INRA, Collège de France, CEREGE, 13545 Aix-en-Provence, France
[2]Institut de physique du globe de Paris, Université de Paris, CNRS, 75005 Paris, France
[3]Laboratoire des Sciences du Climat et de l'Environnement, LSCE/IPSL, CEA-CNRS-UVSQ,
Université Paris-Saclay, 91191 Gif-sur-Yvette, France
[4]Senckenberg Natural History Collections Dresden, 01109 Dresden, Germany
[5]Géosciences Rennes, UMR CNRS 6118, Université de Rennes, 35042 Rennes, France
[6]Institute of Geosciences, Potsdam University, 14469 Potsdam, Germany

**Correspondence:** Agathe Toumoulin (agathe.toumoulin@gmail.com)

**Abstract.** CE1 At the junction of greenhouse and icehouse climate states, the Eocene–Oligocene Transition (EOT) is a key moment in Cenozoic climate history. While it is associated with severe extinctions and biodiversity turnovers on land, the role of terrestrial climate evolution remains poorly resolved, especially the associated changes in seasonality. Some paleobotanical and geochemical continental records in parts of the Northern Hemisphere suggest the EOT is associated with a marked cooling in winter, leading to the development of more pronounced seasons (i.e., an increase in the mean annual range of temperature, MATR). However, the MATR increase has been barely studied by climate models and large uncertainties remain on its origin, geographical extent and impact. In order to better understand and describe temperature seasonality changes between the middle Eocene and the early Oligocene, we use the Earth system model IPSL-CM5A2 and a set of simulations reconstructing the EOT through three major climate forcings: $p\text{CO}_2$ decrease (1120, 840 and 560 ppm), the Antarctic ice-sheet (AIS) formation and the associated sea-level decrease. Our simulations suggest that $p\text{CO}_2$ lowering alone is not sufficient to explain the seasonality evolution described by the data through the EOT but rather that the combined effects of $p\text{CO}_2$, AIS formation and increased continentality provide the best data–model agreement. $p\text{CO}_2$ decrease induces a zonal pattern with alternating increasing and decreasing seasonality bands particularly strong in the northern high latitudes (up to 8 °C MATR increase) due to sea-ice and surface albedo feedback. Conversely, the onset of the AIS is responsible for a more constant surface albedo yearly, which leads to a strong decrease in seasonality in the southern midlatitudes to high latitudes (> 40° S). Finally, continental areas that emerged due to the sea-level lowering cause the largest increase in seasonality and explain most of the global heterogeneity in MATR changes (ΔMATR) patterns. The ΔMATR patterns we reconstruct are generally consistent with the variability of the EOT biotic crisis intensity across the Northern Hemisphere and provide insights on their underlying mechanisms.

## 1 Introduction

### 1.1 Context and aim of the study

The Eocene–Oligocene Transition (EOT) is marked by an abrupt cooling event (∼ 2.9 °C from marine proxies; Hutchinson et al., 2021), regarded as the hinge between the Eocene greenhouse and the later Cenozoic icehouse.

This event is associated with the first major expansion of the Antarctic ice sheet with an estimated sea-level drop of $\sim 70$ m (Hutchinson et al., 2021; Miller et al., 2020). The EOT is described as a relatively brief event ($\sim 790\,000$ years), with two successive steps (at ca. 33.9 and 33.7) recognized in extensively studied marine environments, especially from deep ocean $\delta^{18}$O values (e.g., Katz et al., 2008; Zachos et al., 2001; see the review of Hutchinson et al., 2021).

Reported vegetation responses to the EOT appear to be heterogeneous across continents with important composition changes in some areas (e.g., the west coast of the United States, Greenland, Asia), notably where faunal turnover is important (e.g., Barbolini et al., 2020; Hutchinson et al., 2021; Pound and Salzmann, 2017; Wolfe, 1994). A number of paleobotanical and geochemical studies consistently suggest that the decrease in continental temperatures was particularly marked during winter months, thus leading to higher seasonal temperature contrasts, which are designated as a potential driving mechanism for biotic turnovers (e.g., Eldrett et al., 2009; Mosbrugger et al., 2005; Page et al., 2020; Utescher et al., 2015; Zanazzi et al., 2015). Evoked forcing mechanisms explaining this enhanced winter cooling are $p$CO$_2$, Antarctic ice-sheet (AIS) CE2 inception and increased continentality, although it remains difficult to quantify and disentangle their respective contribution from field data only. Understanding the drivers of these seasonal changes is thus important, not only for assessing the climate system behavior under major $p$CO$_2$ variations but also to better describe the paleoenvironmental context associated with major extinction events of the EOT such as the Grande Coupure in Europe and the Mongolian Remodeling in central Asia (Meng and McKenna, 1998; Stehlin, 1909; see Coxall and Pearson, 2007, for a review).

By comparing paleoclimate simulations to a synthesis of indicators of seasonality changes (Table S1 in the Supplement), our study attempts to retrieve the evolution of seasonal temperature contrast from the middle Eocene to the early Oligocene. The EOT is reconstructed step by step from five simulations, describing the evolution of three major forcings of this time: the $p$CO$_2$ drawdown, the AIS expansion and the resulting sea-level lowering, in order to understand the respective contribution of each component to the resulting seasonality change patterns, along with their possible synergies and retroactions.

## 1.2 Temperature seasonality and its evolution

Temperature seasonality can be quantified by the mean annual temperature range (MATR), which consists of the temperature difference between the warmest and coldest months of the year. Increasing MATR can occur through increased summer temperatures, lowered winter temperatures or both. MATR is practical because it can be directly calculated from temperature proxies providing an estimation of the lowest and highest temperatures within a year. This is the case for the Climate Leaf Analysis Multivariate Program (CLAMP), which reconstructs temperatures from the modern correlation between climate variables and leaf physiognomy (Wolfe, 1993; Yang et al., 2011), or the Coexistence Approach (CA), which uses modern relatives of fossil species to define a mutual climate range of environmental characteristics (Grimm and Potts, 2016; Mosbrugger and Utescher, 1997; Utescher et al., 2014). MATR can also be deduced from the variability of the temperature signal in geochemical proxies for temperatures, stable oxygen isotopes ($\delta^{18}$O), as the time resolution of the proxy rarely allows for the direct reconstruction of seasonal temperatures (e.g., Ivany et al., 2000; Wade et al., 2012).

The spatial distribution of temperature seasonality changes across the EOT appears to be heterogeneous in proxy data (e.g., Pound and Salzmann, 2017). Most changes are described in the Northern Hemisphere from paleobotanical reconstructions and converge to show seasonalities stronger in the early Oligocene than in the mid- to late Eocene. In North America, western and central Europe, seasonality increase is recorded by the decline in species characteristic of warm paratropical to temperate environments such as palms (e.g., *Nypa* sp.), plants from the myrtle and eucalyptus family (Myrtaceae, e.g., *Rhodomyrtophyllum* sp.), conifers (e.g., *Doliostrobus* sp.) and some plant families with tropical elements (e.g., Annonaceae, Lauraceae, Cornaceae, Icacinaceae, Menispermaceae), and, depending on the bioclimatic zones, the expansion of temperate to boreal vegetation through the increase in deciduous and/or coniferous species (Kunzmann et al., 2016; Kvaček, 2010; Kvaček et al., 2014; Mosbrugger et al., 2005; Utescher et al., 2015; Wolfe, 1992). These vegetation changes are associated with a decrease in the coldest month mean temperature (CMMT) across the EOT and start before the EOT at some localities, during the mid- to late Eocene (Moraweck et al., 2019; Mosbrugger et al., 2005; Tanrattana et al., 2020; Tosal et al., 2019; Utescher et al., 2015; Wolfe, 1994). Isotopic analyses have documented this seasonality increase in different continental localities between the Priabonian (37.8 to 33.9 Ma) and the Rupelian (33.9 to 27.82 Ma; Grimes et al., 2005; Hren et al., 2013; Zanazzi et al., 2015). While some of the changes are not directly quantifiable (e.g., the reduction in gastropod growing season length, United Kingdom; Hren et al., 2013; dental morphological changes for grazing perissodactyls, Europe; Joomun et al., 2010), others can demonstrate strong MATR increase (e.g., amplified by 15.6 °C, Canada; Zanazzi et al., 2015). A temperature seasonality increase is also documented for shallow waters of the Gulf of Mexico (increase in the MATR; Ivany et al., 2000; Wade et al., 2012). Some studies have suggested a link between increased temperature seasonality and latitude (e.g., Zanazzi et al., 2007, 2015), but data seem insufficient to validate this relationship, and this trend has not been confirmed by recent palynological compilation (Pound and Salzmann, 2017).

Data from southeast Europe and Anatolia show generally weaker and heterogeneous changes in temperature seasonality, with either no seasonality changes, slight seasonality lowering or slight seasonality strengthening from the mid-Eocene to the Rupelian (Bozukov et al., 2009; Kayseri-Özer, 2013). This variability has been explained by a strong marine influence on this part of Eocene Europe (Kayseri-Özer, 2013). Conversely, north and East Asia temperature seasonality evolution is more comparable to western Europe and North America trends (Quan et al., 2012; Utescher et al., 2015). Vegetation changes reflect an increase in the seasonal temperature range, mainly through the EOT (MATR increase of 2 to 2.5 °C; CMMT decrease of $\sim 2.2$ °C, Quan et al., 2012; Utescher et al., 2015). The appearance of tubers in lotus (*Nelumbo* sp.) during the Eocene suggests the establishment of a dormant phase in these plants and thus of a period unfavorable to plant growth (Li et al., 2014). Fossils showing these structures have been described in southern China (Hainan Province) and easternmost Russia (Kamchatka Peninsula) leading to the hypothesis that they could be favored by cooling and increased seasonality on the East Asian continent during the Eocene (Budantsev, 1997; Li et al., 2014).

In the Southern Hemisphere, studies of Paleogene localities are rarer. Despite a record of late Eocene cooling in Australia, New Zealand and Patagonia, independent proxies (stable isotopes on teeth, bones and pedogenic carbonates, paleobotanical reconstructions) do not suggest a marked temperature seasonality during the Eocene (Colwyn and Hren, 2019; Kohn et al., 2015; Lauretano et al., 2021; Nott and Owen, 1992; Pocknall, 1989). In Australia, the presence of more pronounced tree rings suggests a late Paleogene increase in seasonality starting in the mid-Oligocene at the earliest (Bishop and Bamber, 1985; Nott and Owen, 1992). Finally, the environmental and climatic impact of the EOT in continental Africa remains poorly documented (Hutchinson et al., 2021; Saarinen et al., 2020).

### 1.3 Previous model work

Different modeling studies have illustrated the priming role of $p$CO$_2$ lowering during the EOT, but most focused on oceans through mean annual temperature changes (Baatsen et al., 2020; Goldner et al., 2014; Hutchinson et al., 2018, 2021; Kennedy et al., 2015; Kennedy-Asser et al., 2019, 2020; Ladant et al., 2014b). The model intercomparison study of Hutchinson et al. (2021) has shown a reasonable agreement between modeling experiments with 1120 and 560 ppm (i.e., 4 and 2 times the preindustrial atmospheric levels, respectively, with 1 PAL = 280 ppm and hereafter written "4X" and "2X" CE3) and proxy-data atmospheric and surface ocean temperature reconstructions from the late Eocene and the early Oligocene, respectively. They show, however, that changes in EOT sea surface temperatures (SSTs) were on average best represented by a $p$CO$_2$

shift from 910 to 560 ppm (i.e., a drop of 1.6X). In addition, a recent model–data study of Lauretano et al. (2021) explored Australia climate evolution through the EOT, and estimated a $p$CO$_2$ drop ranging from 260 to 380 ppm (drop of $\sim 0.9$–$1.3X$). A first attempt to explain temperature seasonality change across the EOT was made by Eldrett et al. (2009). In their palynological and modeling study, Eldrett and coauthors explained high-latitude (Greenland) seasonality strengthening by $p$CO$_2$ drop and the consequent increase in sea-ice formation over the Arctic Ocean. In their experiment, sea-ice extension induces a strong albedo feedback, which results in a large decrease in atmospheric temperature during winter. Additionally, changes in geography (topography, land–sea distribution) may have significant effects on terrestrial temperatures at a regional scale (e.g., Lunt et al., 2016; Li et al., 2018). EOT modeling experiments yield mixed answers regarding the temperature feedback resulting from both AIS and contemporary paleogeographic changes (opening of Southern Ocean gateways, Antarctic geography or global geography; Goldner et al., 2014; Hutchinson et al., 2021; Kennedy et al., 2015; Ladant et al., 2014a, b). The few studies testing the combined effect of both AIS and paleogeographic changes (Kennedy et al., 2015; Lauretano et al., 2021) suggested a moderate impact of AIS on global climate sensitivity, as previously suggested by other modeling work (Goldner et al., 2013; Kennedy et al., 2015).

## 2 Material and methods

### 2.1 Model and simulation setting

We used the IPSL-CM5A2 general circulation model, which is built upon the CMIP5 Earth system model developed at the Institut Pierre-Simon Laplace (IPSL), i.e., IPSL-CM5A-LR (Dufresne et al., 2013; Sepulchre et al., 2020). The IPSL-CM5A-LR Earth system model is composed of the LMDZ atmospheric model (Hourdin et al., 2013), the ORCHIDEE land surface and vegetation model (Krinner et al., 2005), and the NEMO v3.6 ocean model which includes modules for ocean dynamics (OPA8.2; Madec, 2016), biochemistry (PISCES; Aumont et al., 2015) and sea ice (LIM2; Fichefet and Morales-Maqueda, 1997). The atmospheric grid has a horizontal resolution of 3.75° longitude per 1.875° latitude (96 × 95 grid points) and is divided into 39 vertical levels. For a more detailed description of the model and its different components, the reader is referred to Sepulchre et al. (2020).

Five simulations were carried out to reconstruct the evolution of temperature seasonality from the middle Eocene to the early Oligocene (Table 1). The applied 40 Ma paleogeography framework is the map developed by Poblete et al. (2021) and already used in Tardif et al. (2020) and Toumoulin et al. (2020). It features common late Eocene geography characteristics such as an open Panama Seaway, an open Tethys with a submerged Arabian Peninsula, a strongly maritime Europe, a Turgai land bridge connecting north-

ern Europe with Asia and narrow Southern Ocean gateways (Fig. 1). The orbital parameters were set to preindustrial values and the solar constant was reduced accordingly to its Eocene value (1360.19 W m$^{-2}$; Gough, 1981). Vegetation was implemented as a boundary condition, using a zonal band of PFTs using modern vegetation distribution patterns.

Simulations were compared in pairs to highlight differences between the middle/late Eocene and the early Oligocene. The simulation set is composed of both realistic and idealized experiments (Table 1). Simulations $4X$, $3X$ and $2X$ represent most of the $p$CO$_2$ range described from the mid-Eocene (Lutetian) to the early Oligocene (Rupelian; Foster et al., 2017). These $p$CO$_2$ values enable the description of the $p$CO$_2$ reduction effect on climate through this time interval and have been used in most previous modeling experiments on the EOT (Hutchinson et al., 2021). Simulations $4X$ and $3X$ cover the range of potential climate values prior to the EOT (Lutetian to Priabonian). The idealized simulation $2X$ allows the identification of a 1 to 2 PAL $p$CO$_2$ lowering alone. In a complementary way, simulations $2X$-ICE and $2X$-ICE-SL describe the early Oligocene climate, following the Antarctic ice-sheet formation. Both simulations are parameterized in the same way apart from the sea level, which is 70 m lower in $2X$-ICE-SL. The use of $2X$-ICE provides a theoretical description of the effect of an ice-covered Antarctica on climate, while $2X$-ICE-SL constitutes a more realistic representation of the early Oligocene climate. In these experiments, the Antarctic ice cap was set to $32.5 \times 10^6$ km$^3$ according to Ladant et al. (2014b). The 70 m sea-level drop was defined following eustatic drop estimates for the EOT (Coxall et al., 2005; Katz et al., 2008; Lear et al., 2008; Miller et al., 2020). It is responsible for important geography changes related to an increase in land proportion, such as the emergence of the Arabian Peninsula and the retreat of the proto-Paratethys epicontinental sea.

All simulations run for 4000 years until temperatures indicate a quasi-equilibrium with only negligible temperature drift within the global mean ocean ($< 0.1$ °C per century; Fig. S1 in the Supplement). These trends are consistent with most model studies and do not affect the quality of atmospheric change described in this study (e.g., Hutchinson et al., 2018; Lunt et al., 2016). The results considered here are averages of the last 100 years of the model runs.

## 2.2 Data compilation

Simulation results were compared to MATR changes ($\Delta$MATR) documented by proxy-data records (Table S1 in the Supplement). We compiled published MATR and CMMT proxy data from various research fields: paleobotany (macrofossils and palynology), geochemistry (isotopic measurements on various material) and paleontology. The data were selected to range from the Lutetian (47.8 Ma) to the end of the Rupelian (27.8 Ma). This large time interval allows the representation of seasonal temperature changes parallel to

the long-term cooling of the Eocene. The inclusion of data from the middle Eocene allows a comparison with simulations testing the effect of a $p$CO$_2$ lowering alone, before AIS formation at the EOT. It is justified by the presence of paleobotanical records suggesting a strengthening of the seasons already from the Lutetian to the Priabonian (e.g., Li et al., 2014; Mosbrugger et al., 2005). Compiled Eocene–Oligocene $\Delta$MATRs correspond either to the values given in original publications, when they were available and precise, or to values recalculated from the original data. For publications for which $\Delta$MATRs were recalculated, we proceeded by grouping the closest sites (especially in terms of latitudes) and checked that the values obtained were consistent with the authors' original interpretation of the paleoenvironmental context. Half of the data come from the pollen compilation of Pound and Salzmann (2017). A selection was made through this study data to keep (1) the best-dated samples, according to their dating quality indicator (data Q1 to Q3; Pound and Salzmann, 2017), and (2) sites with temperature estimates for the Priabonian and Rupelian or at least one nearby locality that could be compared. No Eocene–Oligocene site was selected for more clarity. In an effort to limit the addition of overly uncertain $\Delta$MATR data, sites with a range of CMMT estimates (CMMT$_{max}$–CMMT$_{min}$) $\geq 10$ °C (either for Priabonian or Rupelian sites) were excluded.

Some previously published seasonality increases were not associated with estimates of $\Delta$MATR because seasonality strengthening was either (1) suggested from other parameters, such as the length of the growing season, which does not allow the calculation of the MATR (Hren et al., 2013), or (2) derived from qualitative data that cannot be specifically associated with temperature values (e.g., morphological changes such as tooth shape or plant tuber appearance; Joomun et al., 2010; Li et al., 2014). These sites are displayed on the maps in orange but are not included in quantitative analyses (Sect. 2.3). In order to better estimate the impact of changes in temperature seasonality, the length of the plant growing season (i.e., the number of months with an average temperature above 10 °C) was recalculated using the formula of Grein et al. (2013) for coexistence approach CE4 data (Table 1). Paleocoordinates for every locality were reconstructed using the online service of Gplates (https://www.gplates.org/, last access: TS1), according to the 40 Ma paleogeography used for the paleoclimate models (Poblete et al., 2021) that essentially follows the plate tectonic reconstruction model of Matthews et al. (2016) with some modifications.

## 2.3 Comparison of model and data $\Delta$MATR

Different analyses were made to evaluate the data–model agreement for temperature seasonality changes from the Priabonian to Rupelian (Table 2). Modeled $\Delta$MATR values were extracted from a 3° longitude by 3° latitude area surrounding each data locality. First, a general agreement per-

https://doi.org/10.5194/cp-18-1-2022

**Table 1.** Experimental design. Abbreviations: AIS – Antarctic ice-sheet volume (Ladant et al., 2014b); %Land – total land surface (millions of square kilometers – $10^6$ km$^2$); MAT – mean annual global 2 m air temperature (°C); SST – sea surface temperature (°C). Simulations with an asterisk constitute realistic middle Eocene (Lutetian/Bartonian) and early Oligocene (Rupelian) scenarios; others are either sensitivity experiments ($2X$, $2X$-ICE) or use the high value of the $pCO_2$ range estimated for the time interval ($4X$).

| Simulation | $pCO_2$ | AIS | %Land (Mkm$^2$) | MAT (°C) | SST (°C) |
|---|---|---|---|---|---|
| $4X$ | 1120 ppm | – | 132.3 | 26.4 | 28.2 |
| $3X$* | 840 ppm | – | 132.3 | 23.7 | 25.9 |
| $2X$ | 560 ppm | – | 132.3 | 20.6 | 23.2 |
| $2X$-ICE | 560 ppm | $32.5 \times 10^6$ km$^3$ | 132.3 | 19.7 | 22.9 |
| $2X$-ICE-SL* | 560 ppm | $32.5 \times 10^6$ km$^3$ | 152.7 | 18.7 | 22.2 |

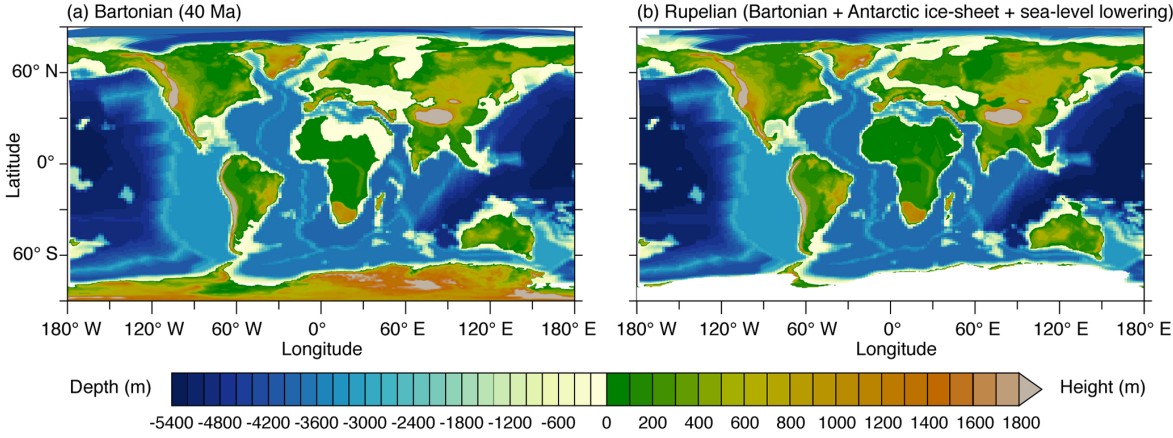

**Figure 1.** Paleotopographic 40 Ma map: **(a)** standard version as used for simulations $4X$, $3X$, $2X$ and $2X$-ICE; **(b)** version adjusted with a homogeneous 70 m sea-level lowering used for the simulation $2X$-ICE-SL (Poblete et al., 2021).

centage was calculated from the direction of seasonality changes alone to assert the agreement between our simulation and qualitative data. For this metric, model predictions are considered "good" for an individual site if the modeled $\Delta$MATR changes in the same direction as the data (i.e., if a modeled $\Delta$MATR increases/decreases at the location of a data point showing seasonality increase/decrease). For data indicating null $\Delta$MATR, good agreement was considered to exist when model values ranged from $-0.5$ to $0.5$ °C.

In addition, Priabonian to Rupelian seasonality changes were compared to model predictions, by (1) testing their correlation and (2) calculating the root of the mean squared distance between their values. These two analyses were performed using R (version 4.0.3; R CoreTeam, 2020, Boston, USA). Given the limited number of quantitative Priabonian–Rupelian data ($n = 29$), the statistical correlation of data–model $\Delta$MATR was assessed from average $\Delta$MATR with the non-parametric Spearman rank test. In this analysis, we used the common significance level, $\alpha$, of 0.05 (i.e., $p$ values $< 0.05$ indicate significant correlations). The root mean squared estimate consists of calculating the root of the mean squared distance between model and data values for comparable points (RMSE; see Kennedy-Asser et al., 2020, and their Fig. S1 in the Supplement for a detailed presentation

of the method). Conversely to the Spearman rank test, for which mean $\Delta$MATR estimations were used, the distance is here measured using the full range of estimates at each data locality (i.e., minimum and maximum $\Delta$MATR). Note that, because it considers the full range of $\Delta$MATR, this method tends to minimize the difference between model and data. The lower bound of modeled $\Delta$MATR at each locality was calculated as the difference between the lowest MATR value over the $3° \times 3°$ area centered around the locality for an Oligocene-like cold simulation ($2X$, $2X$-ICE or $2X$-ICE-SL) and the highest MATR value over the $3° \times 3°$ area for an Eocene simulation ($4X$ or $3X$). For the upper bound, we used the difference between the higher MATR value over the same area for an Oligocene-like cold simulation and the lower MATR value for an Eocene simulation.

The RMSE adjusted to $\Delta$MATR is written as follows[TS2]:

$$\mathrm{RMSE}_{(\Delta\mathrm{MATR})} = \mathrm{SQRT}\left(\sum\left(\Delta\mathrm{MATR}_{(\mathrm{data})} - \Delta\mathrm{MATR}_{(\mathrm{model})}\right)^2 / n\right), \quad (1)$$

where $\Delta$MATR$_{\mathrm{data}}$ and $\Delta$MATR$_{\mathrm{model}}$ are MATR Priabonian to Rupelian changes estimated by data and model, respectively, and $n$ is the total number of localities.

https://doi.org/10.5194/cp-18-1-2022 Clim. Past, 18, 1–22, 2022

# 3 Results

## 3.1 Simulated response to $p$CO$_2$ lowering

In this section, we compare the simulations $4X$, $3X$ and $2X$ together to describe the effects of $p$CO$_2$ drawdown on climate and provide a range of possible MAT and MATR change intensities. The simulation pair $2X$–$4X$ represents the strongest possible changes, $3X$–$4X$ the weakest changes and $2X$–$3X$ an intermediate scenario (see Sect. 2.1). Mean annual temperatures decrease strongly in our different experiments (Table 1, Fig. 2). The halving of $p$CO$_2$ from $4X$ to $2X$ alone (i.e., without AIS formation and sea-level drop) induces a global cooling of 5.8 and 5.0 °C for the air temperature and the surface ocean, respectively (Table 1). A $p$CO$_2$ drop of 1 PAL induces a 2.7 to 3.1 °C lowering of MAT and a 2.3 to 2.7 °C cooling of the SST, for $4X$ to $3X$ and $3X$ to $2X$ changes, respectively. Along with its effect on annual temperatures and regardless of its intensity, $p$CO$_2$ decrease induces zonal ΔMATR including (1) an increase in MATR at high latitudes (especially in the north), (2) a decrease in MATR across most midlatitudes and (3) moderate changes at low latitudes, which we detail in the following Sects. 3.1.2 and 3.1.3 (for Eocene MATR values, see Fig. S2).

### 3.1.1 Areas with increased seasonality

Temperature changes are characterized by polar amplification, with a stronger winter cooling at high latitudes (Fig. 2a, b, e, f, i and j), likely due to the combined effect of albedo and sea-ice feedback. Below a given threshold (situated between 2 and 3 PAL), the $p$CO$_2$ drop enables sea-ice growth over the Arctic and, to a lesser extent, the subsistence of snow on the ground during the cold season, which increases winter surface albedo (Fig. 3). In addition, seasonal sea-ice expansion limits ocean-to-air heat transfer at the highest northern latitudes and contributes to further winter cooling of the atmosphere. This preferential lowering of winter temperatures results in a large MATR increase of 5–20 % ($3X$–$4X$ and $2X$–$3X$) and up to 40 % ($2X$–$4X$; Figs. 4e and S3b) over high northern latitudes. Furthermore, the areas of colder winters and high MATR widen as $p$CO$_2$ decreases: MATR increases from 60° N poleward between $4X$ and 3X, to 50° N poleward between $4X$ and $2X$ (Figs. 4 and S4). In contrast, Antarctica shows moderate ΔMATR (regionally up to 3 °C from $4X$ to $3X$ and 6 °C from 3X to $2X$) compared to high northern latitude lands (6 °C MATR increase from $4X$ to $3X$ and 10 °C from $3X$ to $2X$). This is because Antarctica is characterized by high MATR values in all ice-sheet-free experiments ($4X$, $3X$ and $2X$), resulting in low ΔMATR changes from one experiment to another (Fig. 4a and b). This important seasonality is induced by the continent's high albedo variability, as it oscillates from snow-free to snow-covered soil within a year.

### 3.1.2 Areas with decreased seasonality

Areas with decreased seasonality are characterized by summer cooling that exceeds winter cooling, which reduces the MATR (Figs. 2 and 4). The widest zones of decreasing MATR are continental regions located within the 30–50° latitudinal band. The magnitude of simulated MATR reduction depends on the $p$CO$_2$ drop considered, with a reduction from $3X$ to $2X$ and from $4X$ to $2X$ resulting in up to 20 % and 30 % of regional MATR decrease, respectively (Figs. 4 and S3). At lower latitudes, in Amazonia, equatorial Africa and India, seasonality decreases but to a lesser extent (Fig. 4). A variety of atmospheric and oceanic processes are likely involved in these contrasting MATR changes, depending on the region considered.

The good correlation between MATR increase (respectively, decrease) and TS4 $P/E$ ratio (i.e., precipitation − evaporation, also referred to as *net precipitation*) decrease (respectively, increase; i.e., North America, Central Asia and northern Australia) suggests a strong involvement of the hydrological cycle in this phenomenon (Figs. 5 and 6a, c and h). The $p$CO$_2$ drop tends to slow down the hydrological cycle, which results in flatter $P/E$ latitudinal gradients. At high latitudes, a reduction in precipitation leads to an overall $P/E$ decrease, while at midlatitudes to low latitudes, increased precipitation results in a general increase in $P/E$ (Figs. 5 and 6). In these low-latitude to midlatitude zones, precipitation strengthening is more important in summer and associated with increased evaporation, which results in larger latent heat fluxes and thus in a greater cooling during summer, consistently with decreased seasonality (Fig. 6a, c, h). In addition, summer cooling is strengthened through vegetation feedback: $P/E$ increase favors net primary productivity which in turn contributes to evaporation and summer warmth loss (Fig. 6a, c, h). In contrast, decreased MATR in Europe and southern South America appears poorly correlated to the above-mentioned parameters (Fig. 6b and g). For Europe, the presence of sea ice over the Arctic Ocean (Fig. 3b–e) limits heat loss via the atmosphere during winter and results in a greater summer cooling of the SST (Fig. S5), which contributes to lowering European MATR. In addition, a regional increase in low-level cloud cover during summer could also contribute to lowering CE5 ΔMATR for both Europe and southern South America through albedo feedback (5 % to 15 % higher low-level cloud fraction between 40–60°; Figs. 7 and S6). For southern South America, several parameters seem consistent with the MATR reduction, but it is difficult to disentangle their contribution. By amplifying the latitudinal temperature gradients, the $p$CO$_2$ drop induces a northward migration of the westerly wind maximum (by about 2° of latitude annually but less markedly during austral winter, JAS) and of the Antarctic Circumpolar Current, which delimits the Southern Hemisphere subpolar and subtropical gyres. This northward shift limits the arrival of warm subtropical waters near CE6 the poles (Fig. S7). This greater

Clim. Past, 18, 1–22, 2022                                    https://doi.org/10.5194/cp-18-1-2022

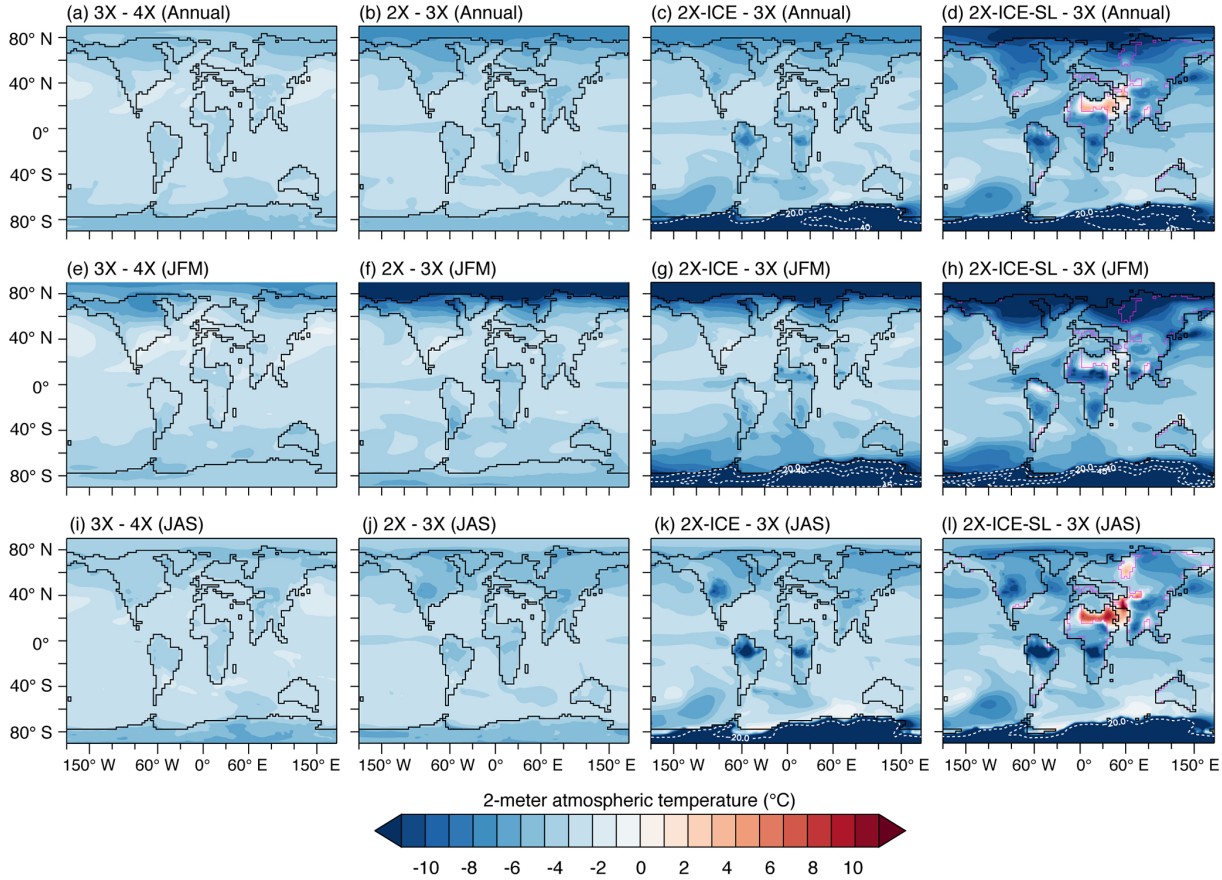

**Figure 2.** Two-meter air temperature changes (°C). JFM: averaged over January to March; JAS: averaged over July to September. Magenta lines of **(d)**, **(h)** and **(l)** indicate shorelines before sea-level lowering. White dotted lines in **(c)**, **(d)**, **(g)**, **(h)**, **(k)** and **(l)** are the level lines encircling the 20, 40 and 45 °C cooling zones.

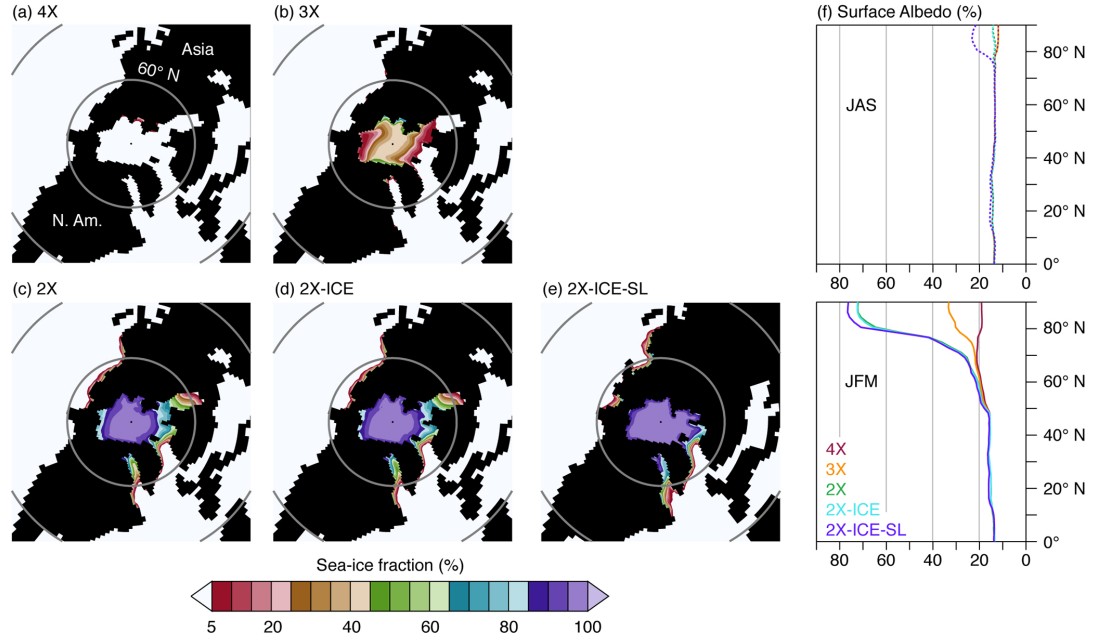

**Figure 3.** Northern Hemisphere winter sea-ice fraction and surface albedo (%).

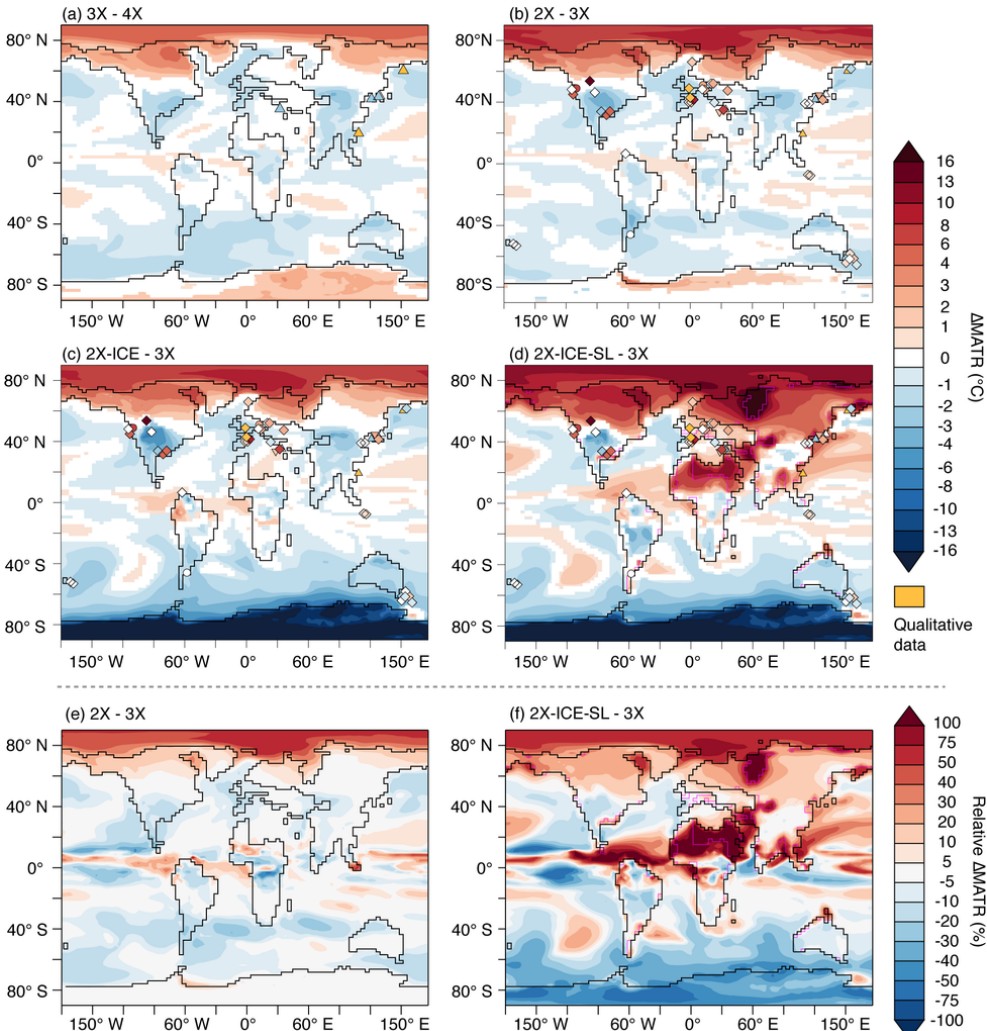

**Figure 4.** TS3 Changes in mean annual temperature range, ΔMATR (°C). Shadings are model differences calculated with a Student $t$ test over the last 100 years of comparative simulations (95 % confidence); white areas indicate no significant ΔMATR. Panels **(e)** and **(f)** indicate relative ΔMATR (in %) for $2X$–$3X$ and $2X$-ICE-SL–$3X$, respectively. Symbols correspond to ΔMATR from proxy data for different time steps: Priabonian–Lutetian (triangles); Rupelian–Lutetian (inverted triangles); Rupelian–Bartonian (circles); Rupelian–Priabonian (diamonds). Orange symbols indicate qualitative values describing a temperature seasonality increase. In the case of proxies reconstructing a range of equally probable values (e.g., the coexistence approach), the values shown are mean values. References are displayed in Fig. 8 and available in the data compilation provided in Table S1 in the Supplement.

cooling in summer SST reinforces the ocean's buffering effect on atmospheric temperatures in southern South America and favors milder summers, and to a lesser extent, cooler winters, which is consistent with a decrease in seasonality (Figs. 4 and 7). Finally, changes in atmospheric dynamics (decrease in the width and increase in the intensity of the Hadley cell) are also visible and could have an impact on air–ocean exchanges, but much more analysis would be needed to understand their implication, which is not the focus of this paper (Fig. S8).

## 3.2 Modeled response to Antarctic ice sheet and sea-level drop

### 3.2.1 Antarctic ice sheet only

AIS formation is responsible for a supplementary 0.9 and 0.3 °C cooling of the air temperature and the surface ocean, respectively (Table 1). Its direct effect on 2 m mean annual temperatures varies regionally and is more striking over Antarctica with up to −35 °C cooling and over the Southern Ocean and Australia (Figs. 2c, g, k and S3c and d). In contrast with Arctic sea ice which increases seasonality at the highest northern latitudes, the AIS decreases southern latitude temperature seasonality (Figs. 4c and S3c and d). Indeed, simu-

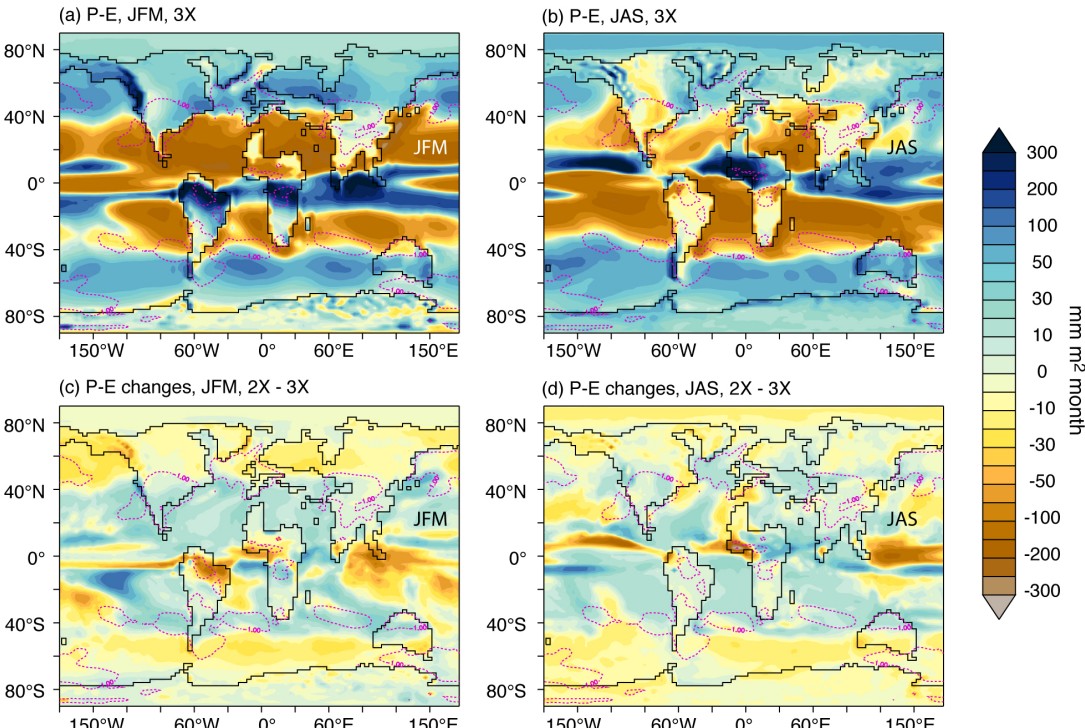

**Figure 5.** Net precipitation (precipitation − evaporation) in JAS and JFM for the late Eocene simulation $3X$ **(a, b)**. Changes associated with $p\mathrm{CO}_2$ drop from $3X$ to $2X$ (differences shown are $2X-3X$) for net precipitation **(c, d)**. Magenta dashed lines contour areas with decreased seasonality ($\Delta$MATR $\leq -1\,°\mathrm{C}$) in $2X-3X$ simulations (blue zones in Fig. 4b).

lations with the AIS have a yearlong white Antarctica with high and stable albedo, which reduces seasonal temperature variability (Fig. 3g–l).

### 3.2.2 Sea-level drop

Sea-level decrease alone is responsible for a 1.0 °C cooling of global MAT (0.7 °C for surface oceans) and results in considerable regional temperature changes in areas with important land–sea distribution changes (Table 1, Figs. 4d and f and S3e and f). The marine regions that become exposed after sea-level drop show the strongest increase in MATR, as they experience both winter cooling and summer warming, due to the lower thermal inertia of land compared to ocean. This seasonality strengthening in newly exposed areas occurs independently of their latitude, therefore disrupting the otherwise zonal distribution of seasonal temperature changes generated by $p\mathrm{CO}_2$ drop and AIS formation (Figs. 4f and S3f). The effect on seasonality of these disappearing seas expands beyond areas adjoining the emerging landmasses due to the resulting regional perturbation in temperature. Northern Africa, western Asia and Russia are the most impacted areas, due to the retreat of the proto-Parathethys sea and the emergence of the Arabian Peninsula (Fig. 1). More moderate seasonality changes are also visible as a result of sea retreats of smaller extent, such as the emergence of the Florida

platform and the modification of the East Asian coastlines (Figs. 4d and S3e and f).

### 3.3 Model–data comparison

#### 3.3.1 $p\mathrm{CO}_2$ lowering

The $\Delta$MATR described by the $p\mathrm{CO}_2$ drop experiments (from $3X$ or $4X$ to $2X$) show no good agreement either with middle to late Eocene or with late Eocene to early Oligocene data estimates (Figs. 4a and b and 8b). The simulations predict no change, or a MATR decrease, in areas where the Lutetian–Priabonian data points ($n = 6$, triangles, Fig. 4a) describe increased seasonality (Fig. 4a and b). Priabonian–Rupelian $\Delta$MATR modeled through $4X$ to $2X$ and $3X$ to $2X$ $p\mathrm{CO}_2$ drops is, on average, lower than data estimates at similar localities, with a mean offset of $-3.5$ and $-2.8\,°\mathrm{C}$, respectively (Fig. 8b; Table 2). The use of the simulation $4X$ instead of $3X$ for the late Eocene stage has a marginal effect on the percentage of agreement for the sign of the $\Delta$MATR, although a slightly higher value is observed for $3X$–$2X$ (Table 2, line "%", Figs. 8b and S9). In addition, the $p\mathrm{CO}_2$ drop alone leads to zonal $\Delta$MATRs which do not transcribe the spatial heterogeneity visible in data. This misfit is visible through high RMSE scores and the absence of significant correlation between modeled $\Delta$MATR resulting from the $p\mathrm{CO}_2$ drop (simulations $2X$–$4X$ and $2X$–$3X$) and $\Delta$MATR described by

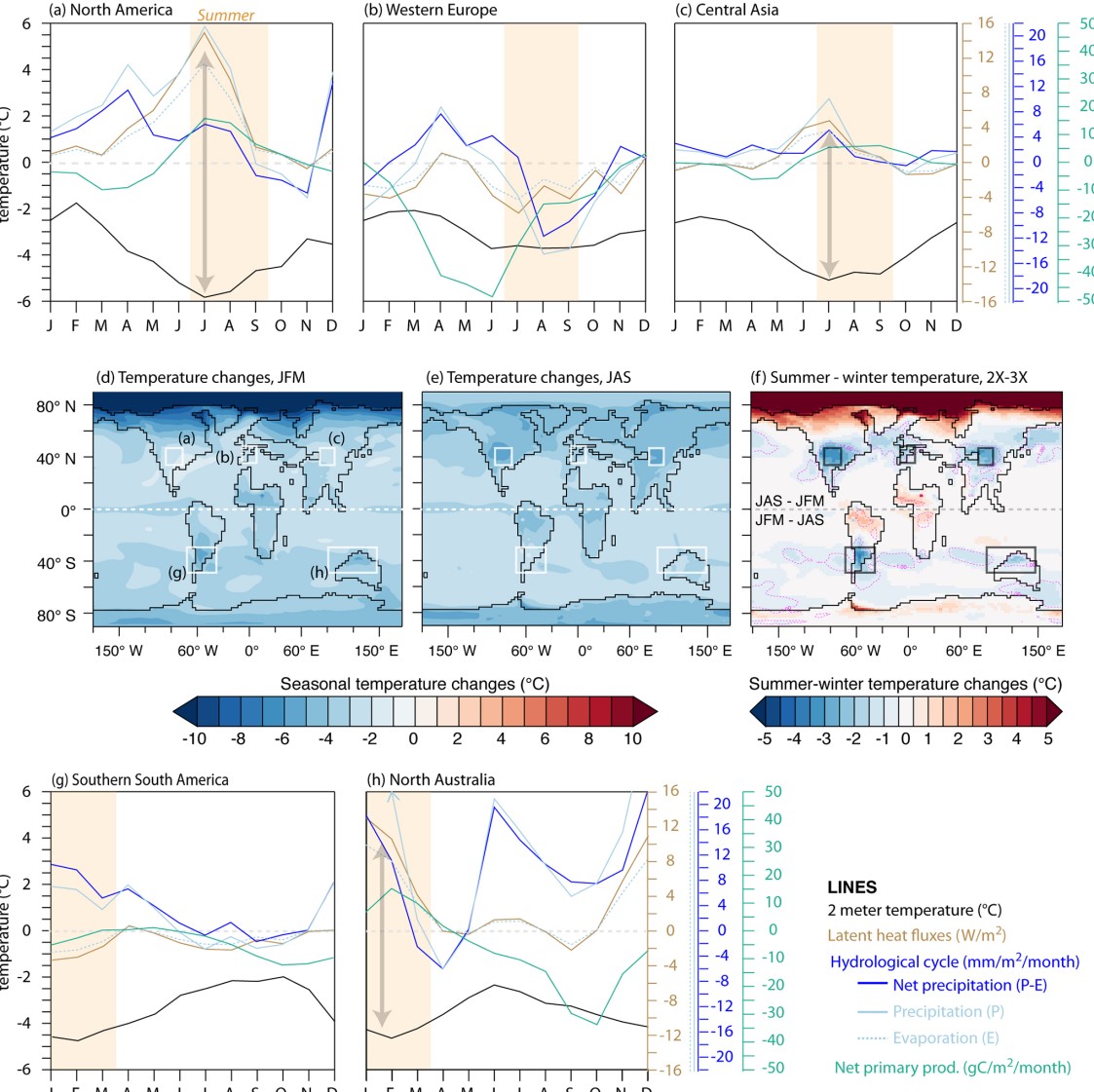

**Figure 6.** Annual variability of multiple climate parameters within the different areas of decreasing seasonality between $3X$ and $2X$ (**a–c**, **g**, **h**): atmospheric temperature (black), latent heat fluxes (soil to atmosphere; brown), hydrological cycle (including precipitation, evaporation and net precipitation; different shades of blue) and net primary productivity (green). Temperature changes and summer−winter temperature changes (**d–f**). Rectangles outline the land areas analyzed in (**a**)–(**c**), (**g**) and (**h**) (ocean zones are not taken into account in the calculation of the plots). Gray arrows show where decreasing summer temperatures match increased latent heat fluxes.

proxy data (Table 2). Two data–model agreement patterns are nevertheless to be noted: (1) regardless of their values (which are higher in data than in our simulations), the northernmost data points are inside or surround the high-latitude seasonality strengthening zone we modeled (Fig. 4, data points 1, 2 and 3 in Fig. 8a); (2) none of the Southern Hemisphere data localities showing no seasonality change are located within MATR increase zones (Fig. 4b).

### 3.3.2 Antarctic ice sheet and sea level

The formation of the AIS alone does not result in a better agreement between modeled and Priabonian–Rupelian ΔMATR estimates. It is even slightly reduced (Table 2). The reinforcement of the MATR lowering zone at high southern latitudes increases the data–model misfit because of data points indicating null ΔMATR in this zone (Figs. 4c and 8b). There is still no significant correlation between ΔMATR from the model and differences documented by proxy data (Table 2).

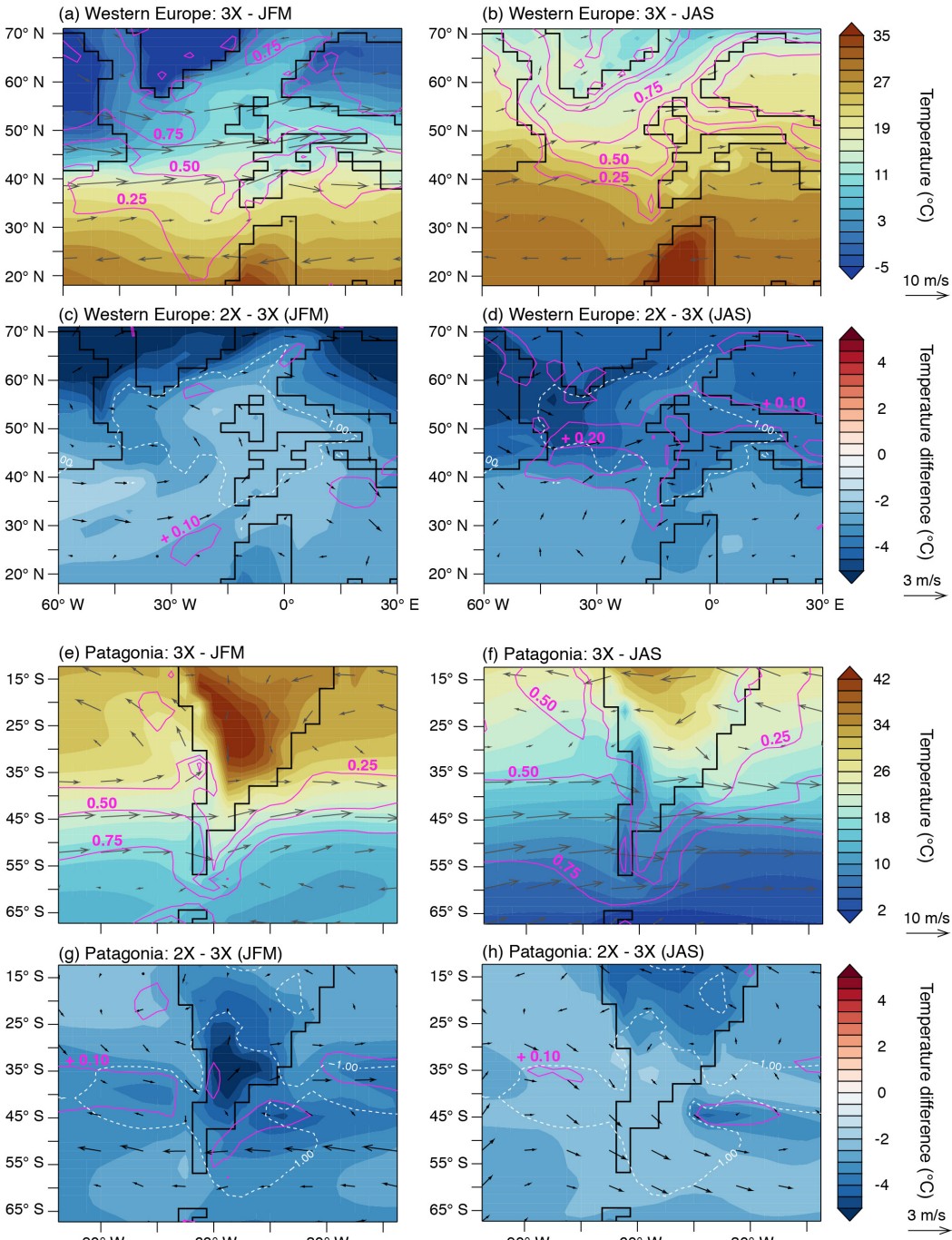

**Figure 7.** Two-meter atmospheric temperature across western Europe and Patagonia. Shadings correspond to temperatures (**a, b, e, f**) and temperature differences (**c, d, g, h**). Similarly, magenta lines contour low-level fraction (**a, b, e, f**) and low-level cloud fraction changes (**c, d, g, h**; always expressed in %) and arrows, 850 hPa winds (**a, b, e, f**) and 850 hPa wind changes (**c, d, g, h**; always in m s$^{-1}$). White dashed lines contour areas with decreased seasonality ($\Delta$MATR $\leq -1\,^{\circ}$C) in $2X-3X$ simulations (blue zones in Fig. 4b).

Geographic changes associated with sea-level drop result in a better agreement with data $\Delta$MATR (Figs. 4d and 8b). The largest continental fraction changes affect the MATR on a broad geographic scale and allow for a better agreement, even with several data points standing away from the

regions directly impacted by the sea-level drop, as for example data estimates located on the Pacific Coast (Fig. 4d). Similarly, coastline changes along the eastern part of Africa and western Asia cause an increase in seasonality in Anatolia and central/western Europe, improving the fit between model

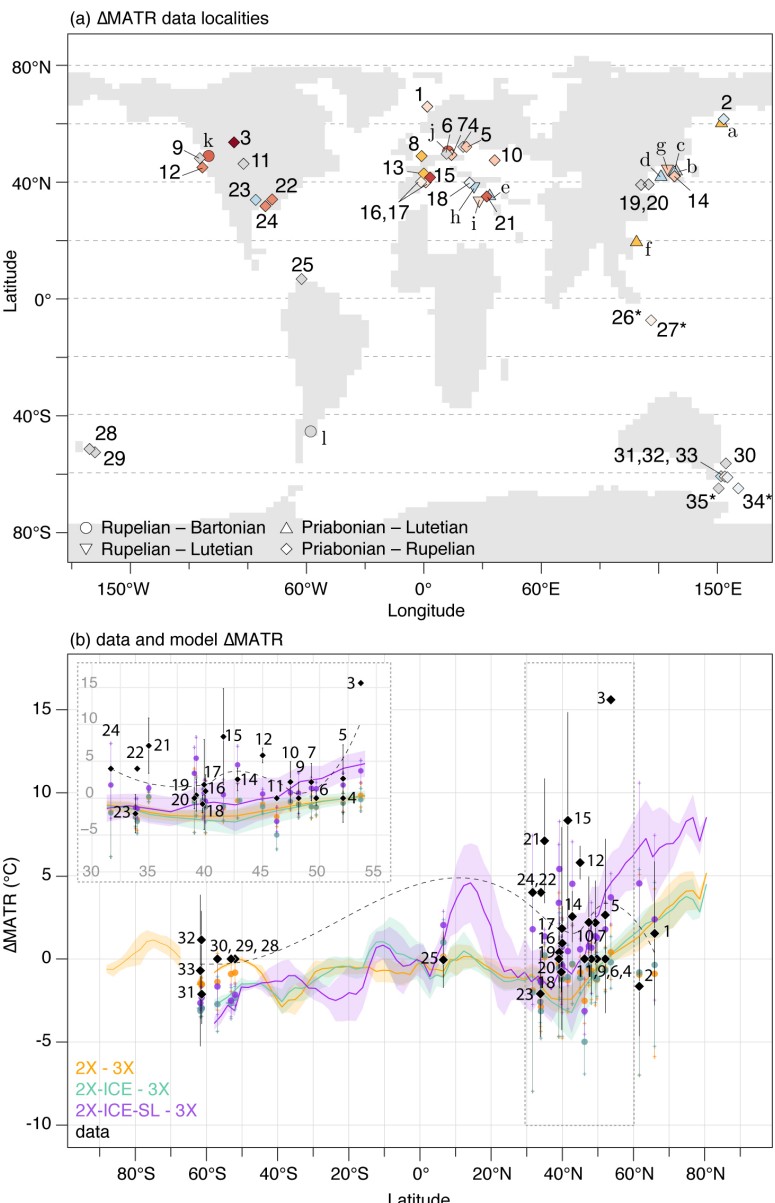

**Figure 8.** Data–model comparison of ΔMATR from the Priabonian to the Rupelian. **(a)** Map of all data ΔMATR estimates compiled in this study (symbols refer to the time period compared for the calculation of the MATR shift; see Table S1 in the Supplement for associated references). The points marked with an asterisk have an uncertain origin (marine sediments) and have not been taken into account in the statistical analyses although they are shown here. The points marked with an asterisk have an uncertain origin (marine sediments) and have not been taken into account in the statistical analyses although they are shown here. **(b)** Comparison of data estimates of Priabonian–Rupelian ΔMATR (black diamonds) to modeled ΔMATR at the same localities (colored circles, calculated over a $3 \times 3°$ area) from different pairs of simulations with $3X$. Error bars are minimum and maximum data estimates of ΔMATR. The dashed black line is the LOESS curve associated with data ΔMATR estimates. Bold colored lines indicate the continental latitudinal gradient of ΔMATR on land (i.e., all longitudes averaged per degree of latitude); color-shaded intervals are the standard deviation around the average. A subfigure similar to **(b)** but using the simulation $4X$ as a Priabonian stage is available in the Supplement, Fig. S9.

and data. These changes in temperature seasonality result in a reduction of 2 to 2.5 months in the duration of the plant growing season (as reconstructed with the formula of Grein et al., 2013; Table S1 in the Supplement). Smaller changes in coastlines such as in Florida, the Kamchatka Peninsula or along the East Asian coast increase seasonality at a regional scale and improve the data–model fit (data points 14, 22 and 24). This better fit is transcribed through the RMSE analysis results, the lowest values being obtained when the simulation $2X$-ICE-SL is used to simulate the Rupelian stage (Ta-

**Table 2.** Priabonian–Rupelian data–model comparison. Negative mean $\Delta$MATR values reflect a tendency of the model to underestimate $\Delta$MATRs. The line "%" gives the percentage of sites where the direction of $\Delta$MATR is adequately modeled (e.g., the model described a MATR reinforcement in the zone where data indicate MATR increase). Modeled $\Delta$MATR estimates were regarded as positive when $> 0.5$, negative when $< 0.5$, or null when in the range of $[-0.5; 0.5]$. $\rho$ indicates the strength of the correlation estimated with the Spearman rank test, with associated $p$ values ($p$). Note that with all $p$ values being $> 0.05$, none of the correlations are significant.

| | 2$X$–4$X$ | 2$X$-ICE–4$X$ | 2$X$-ICE-SL–4$X$ | 2$X$–3$X$ | 2$X$-ICE–3$X$ | 2$X$-ICE-SL–3$X$ |
|---|---|---|---|---|---|---|
| Average $\Delta$MATR model–data mismatch | $-3.5\,°C$ | $-3.9\,°C$ | $-1.9\,°C$ | $-2.8\,°C$ | $-3.2\,°C$ | $-1.2\,°C$ |
| RMSE | $3.1\,°C$ | $3.4\,°C$ | $2.5\,°C$ | $2.9\,°C$ | $3.2\,°C$ | $2.4\,°C$ |
| % | 19.4 % | 19.4 % | 41.9 % | 22.6 % | 16.1 % | 45.2 % |
| $\rho$ | 0.21 ($p = 0.28$) | 0.27 ($p = 0.16$) | 0.29 ($p = 0.12$) | 0.19 ($p = 0.32$) | 0.25 ($p = 0.20$) | 0.29 ($p = 0.12$) |

ble 2). However, there is no significant correlation between model and proxy-data Priabonian–Rupelian $\Delta$MATR, independently of the late Eocene simulation (4$X$ or 3$X$) used as initial stage (for both, Spearman rank test: $\rho = 0.29$, $p$ value $= 0.12$, Table 2). This persistent mismatch may be triggered by biases from the model or the data and from the methodology used to calculate these $\Delta$MATR, which are further detailed in the Discussion. $\Delta$MATR is still slightly underestimated by the model (Fig. 8b; Table 2). The pair of simulations that best describe the Priabonian–Rupelian transition, according to currently available data is 2$X$-ICE-SL–3$X$ as it presents (1) the lowest average model–data $\Delta$MATR mismatch ($-1.20\,°C$) and (2) the best agreement in $\Delta$MATR direction (45 % of the data, Table 2).

## 4 Discussion

### 4.1 Implication for mechanisms of late Eocene to early Oligocene seasonality changes

#### 4.1.1 Model climate sensitivity and climate response to EOT forcing

$\Delta$MATR across the EOT are better predicted when considering the changes occurring between the lower $p$CO$_2$ simulation 3$X$ for the late Eocene stage and the most realistic simulation 2$X$-ICE-SL for the early Oligocene stage. In addition, with a mean SST cooling of 2.7 °C between 3$X$ and 2$X$ simulations (Table 1), surface temperature changes are also in agreement with the mean changes described in marine proxies across the EOT (i.e., difference of 2.9 °C between 38–34.2 and 33.7–30 Ma, Hutchinson et al., 2021). This best fit with a limited drop in $p$CO$_2$ reflects the high climate sensitivity of our model (i.e., the average temperature change per doubling of the $p$CO$_2$ at model equilibrium; PALAEOSENS, 2012). This high sensitivity is also highlighted in our experiments of $p$CO$_2$ halving from 1120 to 560 ppm (2$X$–

4$X$), which result in a dramatic mean annual global cooling (5.8 °C for global MAT, 5 °C for SST, Table 1). Such a temperature difference is high compared to previous modeling studies which describe a 3 to 4 °C surface atmospheric temperature difference under a similar $p$CO$_2$ decrease and the 2.9 °C cooling found in marine proxies across the EOT (Hutchinson et al., 2021). The $p$CO$_2$ of 4$X$ and 2$X$ is commonly CE7 used in simulations to represent the transition to an icehouse world (e.g., Baatsen et al., 2020; Goldner et al., 2014; Kennedy-Asser et al., 2019). Although 4$X$ is likely to represent the upper end of $p$CO$_2$ possible values for the late Eocene (the average values are rather close to 800 ppm from the Lutetian to the Priabonian; Foster et al., 2017), the use of this value is justified to better reconstruct high-latitude temperatures (Huber and Caballero, 2011). A good agreement between warm conditions and Bartonian SST data has also been recently shown by other experiments using the model IPSL-CM5A2 with middle/late Eocene boundary conditions (Tardif et al., 2020; Toumoulin et al., 2020). We thus argue that although the use of the 4$X$ simulation is appropriate to study possible variations in $p$CO$_2$ during the Eocene, the use of the 3$X$ simulation is better to study the changes between the Priabonian and the Rupelian.

#### 4.1.2 Temperature seasonality changes through the late Eocene

The evolution of the different climate features likely involved in $\Delta$MATR is consistent with several findings from previous studies. First, earlier modeling experiments have described albedo and sea-ice increase resulting in polar amplification of the cooling (e.g., Baatsen et al., 2020; Kennedy-Asser et al., 2019) and a reinforcement of temperature seasonality (Eldrett et al., 2009). The resulting strengthening and expansion of the northern high-latitude MATR increase zone with $p$CO$_2$ lowering is a good explanation for the dramatic seasonality increase at high latitudes suggested by some studies

https://doi.org/10.5194/cp-18-1-2022

(Eldrett et al., 2009; Wolfe, 1992; Zanazzi et al., 2015). In addition, changes in the magnitude and distribution of net precipitation (i.e., precipitation – evaporation) resulting from the decrease in $p\text{CO}_2$ agree with previous theoretical and modeling work suggesting an intensified hydrological cycle under higher $p\text{CO}_2$ (e.g., Carmichael et al., 2016; Hutchinson et al., 2018). This phenomenon results from a greater capacity of the air to retain moisture and more intense atmospheric convection phenomena (Allen and Ingram, 2003; Carmichael et al., 2016; Held and Soden, 2006). In parallel, although the implication of changes in the atmospheric circulation in the southern South American seasonality lowering zone appears non-obvious, the intensification and weakening of the Hadley cell extent in relation to changing $p\text{CO}_2$ levels have been described numerous times (e.g., Lu et al., 2007; Frierson et al., 2007). Deeper analyses would be needed to understand the atmospheric dynamics in the simulations, which is beyond the scope of the study. Finally, the increase in low cloud cover is consistent with previous model studies describing a higher fraction of low-level clouds under lower $p\text{CO}_2$ (Baatsen et al., 2020; Caballero and Huber, 2013; Zhu et al., 2019). Nevertheless, although a low-level cloud cover increase due to the $p\text{CO}_2$ drop is consistent with increased air moisture in western Europe at the EOT (Kocsis et al., 2014), this parameter remains poorly constrained in paleoclimate archives and modeling analysis (Lunt et al., 2020; Sagoo et al., 2013).

Despite these agreements, the MATR evolution resulting from the $p\text{CO}_2$ drop does not clearly match data estimates, whether they correspond to both Lutetian–Priabonian or to Priabonian–Rupelian changes. This suggests that the temperature seasonality inferred from proxy data can only be partly explained by a $p\text{CO}_2$ drop. Since zonal $\Delta$MATR patterns are simulated with a $p\text{CO}_2$ drop of 1 PAL (either from $4X$ to $3X$ or from $3X$ to $2X$), we hypothesize that they likely occurred before the AIS onset and that the strengthening of seasonality occurred in northern high latitudes in the first place. However, most Lutetian–Priabonian data are not located in the high latitudes, which prevents unambiguous testing of this hypothesis (Fig. 4a and b and Table S1 in the Supplement). Similarly, the presence of areas with decreased seasonality due to changes in the hydrological cycle (i.e., the USA, Central Asia, north Australia) cannot be confirmed because of a lack of data in these areas: although some of the data associated with a decrease in the MATR share the same latitudinal bands, none of them are directly located within a zone of MATR decrease. New additional seasonal temperature records in these areas would be interesting to better trace such eventual early trends. The general low fit of data and model values for middle to late Eocene changes is, to some extent, predictable since the ice-free $2X$ simulation does not represent the late Eocene (see "Material and methods" section) and was designed as a sensitivity test. Indeed, small-scale glaciations (25 %–35 % modern AIS) may have existed during the late Eocene, before the EOT, associated with a moderate sea-level decrease (Carter et al., 2017; Miller et al.,

2020; Scher et al., 2014). Interestingly, the combination of the three forcing mechanisms leads to a better agreement of the modeled $\Delta$MATR with some of the few middle to late Eocene data, especially in coastal areas of Kamchatka, and southern China (triangles, Fig. 4). Although the 70 m sea-level decrease prescribed in the $2X$-ICE-SL simulation is unrealistic for the late Eocene, the better data–model agreement when both AIS and sea-level decrease are considered suggests that small ice-sheet development before the EOT may have played a significant role in driving the middle to late Eocene $\Delta$MATR. Additional sensitivity experiments, with a smaller AIS and an intermediated sea-level drop, may allow further quantification of the sensitivity of coastal localities to sea-level variations occurring before the EOT.

### 4.1.3 Temperature seasonality changes through the EOT

The use of two simulations to set up the effect of the AIS onset (with or without a drop in sea level) is interesting to unravel the direct and indirect mechanisms affecting temperatures. Temperature changes resulting from the presence of the AIS alone (i.e., not taking into account sea level) are consistent with previous model studies that simulate its highly regional effect on atmospheric temperature (see the Supplement of Hutchinson et al., 2021), although the changes in our simulations spread more widely over the Southern Ocean and Australia. The decreasing seasonality zones modeled at high latitudes and midlatitudes of the Southern Hemisphere are mostly associated with an absence of seasonality change in the data, which often display stable vegetation and biomes from the late Eocene to the Rupelian (Hutchinson et al., 2021; Kohn et al., 2015; Nott and Owen, 1992; Pocknall, 1989; Pound and Salzmann, 2017). This apparent mismatch calls into question the capability of paleobotanical proxies to record a temperature seasonality decrease in environments already characterized by low seasonality. Indeed, the decrease in the temperature seasonality is associated with a more pronounced drop in summer temperatures, which is a less limiting factor for flora distribution and thus less constrained in the fossil record than winter temperatures (Huber and Caballero, 2011).

The evolution of temperature seasonality from the Priabonian to the Rupelian is better represented when the sea-level drop associated with the AIS is taken into account (Table 2, Figs. 4 and 8). This consequence of the Antarctic glaciation has global repercussions and explains part of the heterogeneity documented in the data, as previously suggested (Pound and Salzmann, 2017). Our results are very dependent on the paleogeography used in the simulations and of the proxy location used in our data–model comparison. Because our Rupelian simulations use a late Eocene paleogeography with a global sea-level decrease, we overlook some paleogeographic changes that occurred between both periods, which may affect our seasonality reconstruction. The

gradual northward migration of Australia is not considered; the Neotethys is gradually closed during the early Oligocene, but a deep-sea passage to the north of the Arabian Plate remains present in our paleogeography (Barrier et al., 2018). Another source of error may come from fragmented continental areas such as those seen in Europe at that time. In these zones, temperature changes recorded through the EOT are heterogeneous as paleovegetation studies suggest medium (1.8–2.1 °C; Moraweck et al., 2019; Teodoridis and Kvaček, 2015; Tosal et al., 2019) to strong (up to 8.3 °C; Tanrattana et al., 2020) MATR increase. The heterogeneity shown in data might thus result from smaller-scale paleogeographic changes through the EOT that are not well represented by the resolution used in our simulations. This variability of the data $\Delta$MATR estimates could also be due to (1) a variable quality of MATR data related to the fragmentary nature of the fossil record and to differential recording of vegetation types as well as (2) differences in the temperature of marine/oceanic zones before regression. Depending on their extension and depth, these seas may have buffered seasonal temperature variations in the nearby regions more or less importantly, and therefore their disappearance may have affected the MATR at different magnitudes. An early Oligocene intensification of seasonality in central and eastern Europe, associated with a major phase of Antarctic ice-sheet expansion (and its effect on sea level), is consistent with fairly stable vegetation between the middle and late Eocene (Bozukov et al., 2009; Kvaček et al., 2014; Moraweck et al., 2015). This may result from the proximity with the warm Tethys, buffering the EOT cooling, as suggested by stable $\delta^{18}$O describing moderate temperature changes in this area (Kocsis et al., 2014).

Finally, differences between our modeling results and data may also be related to the amplitude of the sea-level drop used in our simulation compared to its variability during the Rupelian. The EOT is generally described in two steps: a first event at $\sim$ 33.9 Ma with both a decrease in temperature and sea level ($\sim$ 25 m) and a second event, the Early Oligocene Glacial Maximum (EOGM), between approximately 33.65 and 33.15 Ma, starting after a large oxygen isotope incursion (often referred to as "Oi-1"), which is characterized by an additional 50 m sea-level decrease (see Hutchinson et al., 2021, for synthesis and terminology, and Miller et al., 2020). The sum of these two steps corresponds to the boundary conditions of our simulation. However, important variations in the East Antarctic Ice Sheet have been described until the early Miocene (50–60 m sea-level equivalent; Miller et al., 2020). Directly after the EOGM phase, a decrease in ice volume is visible between 33.15 and 32.8 Ma, before it increases again and remains stable between 32.8 and 29 Ma (after the Oi-1a event; Galeotti et al., 2016). Due to the combined effects of the drop in $CO_2$ and the development of the AIS (and the amplitude of the associated drop in sea level, 70 m), the important changes in seasonality reconstructed here ($2X$-ICE-SL$-3X$) were probably not in place throughout the Rupelian but rather for shorter periods during the EOGM, or later be-

tween 32.8–29 Ma. Most continental paleoclimate studies do not provide the resolution to distinguish these steps. Among the data points compiled for this study, only four sites have enough temporal resolution to be linked to the EOGM phase represented by our $2X$-ICE-SL simulation (Bozukov et al., 2009; Hren et al., 2013; Kohn et al., 2015; Tosal et al., 2019).

## 4.2 Perspectives on environmental and biotic crisis

The EOT is associated with major extinction events, of which the best known are the Grande Coupure in Europe and the Mongolian Remodeling in central Asia (Stehlin, 1909; Meng and McKenna, 1998; see Coxall and Pearson, 2007, for a review). Both events have been recognized as major biotic turnovers for ungulates (Blondel, 2001; Stehlin, 1909). In addition, other vertebrates were affected by the Grande Coupure (e.g., rodents, primates, amphibians and squamates), and major changes in vegetation are described, in association with the Mongolian Remodeling and regionally, in Europe (e.g., Barbolini et al., 2020; Marigó et al., 2014; Pound and Salzmann, 2017; Rage, 1986, 2013; Roček and Rage, 2003). These changes have been linked to (1) competitive interactions resulting from the dispersal of Asian taxa to Europe and (2) EOT climate deterioration and selection processes through resource and/or habitat changes (e.g., Hooker et al., 2004; Kratz and Geisler, 2010; Marigó et al., 2014; Sun et al., 2015; Zhang et al., 2012). The latter mechanism is commonly related to irreversible cooling and/or aridification at the EOT (e.g., Blondel, 2001; Sun et al., 2015; Zhang et al., 2012). Climate cooling may have significantly reduced the habitat of well-spread early Eocene tropical (and paratropical) species, which are characterized by narrow thermal ecological niches (Hren et al., 2009; Huang et al., 2020; Jaramillo et al., 2006; Wing, 1987). Although the distribution of fauna and flora is based on a complex set of parameters, we discuss here how $\Delta$MATR provides an additional interpretative key for understanding biotic turnover at the EOT.

While North America and Asia show comparable temperature changes, our simulations highlight significant differences in the evolution of their MATR, which decreases and either increases/decreases at a regional scale, respectively. Vegetation changes and the Mongolian Remodeling are contemporaneous to AIS growth between 32.8–29 Ma and can be compared with our $2X$-ICE-SL simulation (Galeotti et al., 2016; Kraatz and Geisler, 2010; Sun et al., 2015). The MATR strengthening modeled in central Asia shows that cooling was particularly strong during winter. In addition to the aridification, this more pronounced winter cooling may have contributed to the intensity of extinctions in this area (Barbolini et al., 2020; Dupont-Nivet et al., 2007). This seasonality strengthening is strongly driven by the proto-Paratethys retreat, which contrasts with a previous geochemistry study suggesting a weak contribution of this sea to local climate conditions (Bougeois et al., 2018). Conversely, the

zone of decreased MATR reconstructed in North America may provide an explanation for the low impact of the EOT on fauna and vegetation in this area (Coxall and Pearson, 2007; Prothero and Heaton, 1996; Stucky, 1992). Despite a similar decrease in the mean annual temperature, most of the temperature drop is in summer and is not associated with the onset of cold winters (Fig. S4). We hypothesize that these patterns enabled a greater persistence of existing warm-temperate to paratropical vegetation and associated biota (Pound and Salzmann, 2017). The study of Tardif et al. (2021) using the same model and similar simulations ($4X$ and $2X$-ICE) with a dynamic vegetation module shows moderate biome changes across the EOT in this North America decreased seasonality zone. Reduced stress on biodiversity in areas with limited or reduced MATR changes is also consistent with moderate vegetation changes across the EOT in areas with decreasing seasonality in the Southern Hemisphere (see discussion in Sect. 4.1.3).

Europe stands in an intermediate position between North America and Asia with generally weaker changes in MATR (Fig. 4d); eastern Europe exhibits a slight increase in MATR, while MATR decreases in the west. Although comparable $\Delta$MATR values could have different impacts depending on initial MATR, the types of ecosystems and their resilience, the strong consequences of moderate seasonality increase on growing season length support the hypothesis that seasonality changes may also have contributed to shaping the biodiversity evolution in central Europe. The late development of increased seasonality zones in this area, linked to the major phase of sea-level drop, could explain the persistence of fairly stable vegetation during the Eocene (Kvaček et al., 2014; Hutchinson et al., 2021). Yet, given the fragmented nature of Europe at this time, increases in seasonality prior to the EOT would also be possible as a result of smaller, but locally significant, sea-level variations. These differences in the evolution of the MATR between North America, Europe and Asia are consistent with several studies suggesting different causes for EOT extinctions (Blondel, 2001; Hooker et al., 2004; Meng and McKenna 1998; Sun et al., 2015). Finally, little is known about the EOT in Africa (notably because of a few Oligocene sites), but the data available suggest moderate changes in northern African flora and fauna (Hutchinson et al., 2021; Jacobs et al., 2010; Pound and Salzmann, 2017; Rasmussen et al., 1992). The significant increase in seasonality in North Africa seen in our simulations (which results from the emergence of part of the continent) differs from the one observed in other areas, since it is linked to an increase in summer temperatures (barely no change in winter). Unchanged winter temperatures and the shift of marine to terrestrial environments might complicate the recording of this temperature seasonality strengthening. Recording low-latitude changes in seasonality strengthening may however be possible as shown by a recent study combining multiple lines of evidences (including plant and primate macrofossils), which suggest a potential shift from tropical to more open deciduous vegetation through the EOT, reflecting increased seasonality in precipitation and/or temperature seasonality during the early Oligocene at low latitudes of South America (Peru, $\sim 7°$ S, Antoine et al., 2021). More studies would probably enable a better understanding of the evolution of the seasonality in this low-latitude area.

## 5   Conclusions

This study investigates the changes in temperature seasonality during the middle to late Eocene and across the EOT. MATR changes modeled with the combined effects of $p$CO$_2$ drop, AIS formation and sea-level lowering are qualitatively consistent with the proxy-data reconstruction of the late Eocene to early Oligocene. The decrease in $p$CO$_2$ leads to a marked strengthening of seasonality in the northern high latitudes, which may have started earlier than the EOT, during the late Eocene. The formation of the AIS and the resulting sea-level drop lead to both an intensification and an extension of increasing seasonality areas. The best agreement between data and modeled MATR evolution throughout the EOT is reached when all three parameters are combined. Accounting for sea-level changes associated with the Antarctic glaciation appears to be the most important parameter to explain the heterogeneity of $\Delta$MATR across the EOT. A seasonality increase is also visible in middle to late Eocene localities, which may reflect earlier sea-level changes associated with the incipient precursors of the Antarctic ice sheet. A discrepancy between data and model is present for quantitative MATR change estimates across the EOT, with less marked seasonality changes in the model. This is mainly due to areas where a decrease in seasonality is predicted by the model while the vegetation proxies show stable vegetation. We hypothesize that this discrepancy can be explained by a low capacity of vegetation to record decreases in summer temperatures compared to winter temperatures.

Reconstructing changes in MATR brings additional constraints on the abiotic environmental factors at play on land between the middle Eocene and the early Oligocene. The different mechanisms described here likely explain the heterogeneity in seasonality changes found in data across the greenhouse–icehouse transition and provide insights into the diversity of continental paleoenvironments. The map of $\Delta$MATR reconstructed here give new elements to help understand major extinction events of the EOT. This study primarily focused on the evolution of temperature seasonality that has the clearest evolution. The variability of other seasonality parameters, including rainfall seasonality, will be worth investigating in future studies to obtain a finer picture of the evolution of terrestrial climates and biodiversity through the EOT. In addition, further work using higher-resolution Rupelian paleogeography and regional models would be of great interest to better reconstruct temperature

seasonality changes, especially in highly fragmented land areas such as Europe during the Eocene.

**Code availability.** LMDZ, XIOS, NEMO and ORCHIDEE are released under the terms of the CeCILL license. OA-SISMCT is released under the terms of the Lesser GNU General Public License (LGPL). The IPSL-CM5A2 code is publicly available through svn, with the following command lines: svn co http://forge.ipsl.jussieu.fr/igcmg/svn/modipsl/branches/publications/IPSLCM5A2.1_11192019 (last access: 16 February 2021, IPSL Climate Modelling Centre, 2021a), modipsl cd modipsl/util;./model IPSLCM5A2.1

The mod.def file provides information regarding the different revisions used, namely

– NEMOGCMbranchnemo_v3_6_STABLErevision666-XIOS2branchs/xios-2.5revision1763

– IOIPSL/srcsvntags/v2_2_2

– LMDZ5branches/IPSLCM5A2.1rev3591

– branches/publications/ORCHIDEE_IPSLCM5A2.1.r5307rev6336–OASIS3-MCT2.0_branch(rev4775IPSLserver).

The login/password combination requested for the first use to download the ORCHIDEE component is anonymous/anonymous. We recommend referring to the project website http://forge.ipsl.jussieu.fr/igcmg_doc/wiki/Doc/Config/IPSLCM5A2 (last access: 16 February 2021, IPSL Climate Modelling Centre, 2021b) for a proper installation and compilation of the environment.

**Data availability.** The key climatological outputs of the simulations are stored in the PANGAEA database: https://doi.pangaea.de/10.1594/PANGAEA.930422 (Toumoulin et al., 2021).

**Supplement.** The supplement related to this article is available online at: https://doi.org/10.5194/cp-18-1-2022-supplement.

**Author contributions.** YD, AT and JBL conducted the modeling experiments. AT compiled proxy data, analyzed the model results, generated figures and tables, and drafted the paper. AT, YD, DT and JBL discussed the paleoclimate results. LK contributed to the paleobotanical context and AL and GDN to the paleogeographic and geological context. All authors have provided critical feedback and contributed to the final paper.

**Competing interests.** The contact author has declared that neither they nor their co-authors have any competing interests.

**Disclaimer.** Publisher's note: Copernicus Publications remains neutral with regard to jurisdictional claims in published maps and institutional affiliations.

**Acknowledgements.** We thank the CEA/CCRT for providing access to the HPC resources of TGCC under the allocation 2018-A0030102212, and 2019-A0050102212 made by GENCI. Agathe Toumoulin and Guillaume Dupont-Nivet acknowledge the support of the ERC MAGIC under grant 649081. Yannick Donnadieu and Delphine Tardif acknowledge support from ANR AMOR (grant no. ANR-16-CE31-0020). The authors acknowledge Cheng-Sen Li for sharing a translation of the reference Budantsev (1997). We acknowledge the use of Ferret (https://ferret.pmel.noaa.gov/Ferret/, last access: 10 November 2021) and RStudio software (https://www.rstudio.com/, last access: 10 November 2021) for analysis and figures in this paper. We sincerely thank Alberto Reyes for his editorial handling and two anonymous reviewers for their interesting comments that improved the quality of this article.

**Financial support.** This research has been supported by ERC MAGIC (grant no. 649081) and ANR AMOR (grant no. ANR-16-CE31-0020). The article processing charges for this open-access publication were covered by ANR AMOR (ANR-16-CE31-0020). CE9

**Review statement.** This paper was edited by Alberto Reyes and reviewed by two anonymous referees.

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

## Remarks from the language copy-editor

**CE1**  You requested a number of stylistic changes, which have been addressed and/or inserted. However, we normally ask that authors restrict their requested changes to necessary corrections and responses to the production marks and refrain from making stylistic changes. This is because changes always run the risk of inadvertently inserting errors and typos and we wish to avoid a corrigendum at a later date. I should be very grateful if you would not send in any more stylistic changes in the second round of proofing. Thank you!

## Remarks from the typesetter