# Peer review of "Evolution of continental temperature seasonality from the Eocene greenhouse to the Oligocene icehouse - A model-data comparison"

_Climate of the Past, 2021_

## Author Comment (AC1)

Montpellier, July 26[th] 2021

Dear reviewer,

We sincerely thank you for reviewing our manuscript and are grateful for your constructive comments recognizing the value of our work. We carefully accounted for all your comments and questions and provided detailed answers here.

The main remarks concerned the possible mechanisms associated with the areas of decreasing temperature seasonality and the presentation of these results with the limitations. To address this, we made new diagnostics which we may include as additional figures in the manuscript.

In addition, while correcting our manuscript, we felt it was unfortunate not to include data from the compilation of Pound and Salzmann (2017), given the small number of data available. We propose to increase the number of data and thus update the results accordingly as presented in a dedicated section after answering your comments. We are aware that this kind of practice is not usual, and we apologize for the extra work it may require, but we believe it will give more representative results of the changes of the Eocene-Oligocene Transition. The message of the paper and the conclusions remain the same.

Overall, we feel the manuscript is greatly improved by these substantial revisions.

Best regards,

Agathe Toumoulin on behalf of all co-authors.

This is my first review of the manuscript by Tourmoulin et al., titled "Evolution of continental temperature seasonality from the Eocene greenhouse to the Oligocene icehouse - A model-data comparison". The authors use a series of model simulations to investigate changes in seasonality (Mean Annual Temperature Range, MATR) across the Eocene-Oligocene transition (EOT). They also compile published estimates of temperature seasonality proxies and compare those to their model simulations. I think the manuscript provides a significant contribution providing a new view on the changes taking place across the EOT. The manuscript is also well written, mostly easy to read, and contains a wealth of background information. My main critique concerns the presentation of the results, as I think the authors could make their arguments stronger with a bit more analysis and/or by better acknowledging the limitations of their study. I suggest a major revision, but they should be quite straightforward to do. After a revision, I believe the manuscript would be worthy of prompt publication in the Climate of the Past. Please find my detailed comments below:

Major comments:

**1 Overall, I think 3.1.3 is cutting corners, it might well be that some of the stated mechanisms are true, but it is difficult (if not impossible) to confirm the mechanisms based on the evidence presented. For this type of paper, it is not crucial to identify the exact mechanism, although it is valuable of course. I would suggest the following 1) to be able to make a bit more robust statement, the authors could check the correlation between surface air temperature change and latent heat flux change/P-E change/Primary prod. Change. 2) I would change the language towards 'we suggest that this phenomenon could be explained by...' rather than 'this phenomenon is well explained by'.**

Thank you for this comment. We recognize that the mechanisms could be better explored. In order to take this comment into account, we have performed additional diagnostics to better evaluate the potential mechanisms at stakes in our simulations. We focused particularly on the regions demonstrating a decreased seasonality in the early Oligocene (mostly at mid-latitudes), given that

they constitute the most counter-intuitive result of this study. To do so, we extracted the anomalies in Precipitations, Evaporation, resulting P-E, NPP and surface temperature on land areas between 2X and 3X simulations (since the areas of decreasing seasonality appear as a result of decreasing $CO_2$, see v1 of the manuscript, see Figure below).

We feel no evident correlation or single mechanism emerges from these diagnostics and the magnitude of these changes is highly variable from one region to the other which supports that decreasing temperature seasonality may result from various mechanisms depending on the considered area. The agreement between decreasing summer temperatures and increasing latent heat fluxes/net precipitation appears particularly good over the United States, and less so over Asia and Australia. In contrast, additional mechanisms are needed to explain the temperature changes over Europe and southern South America (as mentioned in the first version of the manuscript, see also our answer to your next comment). Finer analysis involving perhaps daily to hourly resolution might be necessary to provide a better understanding of the mechanisms at stake

We have restricted Fig. 5 to subfigures (a-d) and made a new figure (see below) showing co-variations between temperature, latent heat, hydrological cycle (precipitation / net precipitation / evaporation), and net primary productivity. We propose to add this Figure to the supplementary material since it provides information on specific climate mechanisms that are not necessary for the understanding of the manuscript.

We modified the text of section 3.1.3 accordingly and added a short sentence in the discussion (section 4.1.2) to discuss the changing extent of atmospheric cells in greenhouse climates. As suggested we also reformulated the sentence originally located l. 263.

New sentence section 4.1.2.:
"In parallel, the intensification and weakening of the Hadley cell extent in relation to changing pCO2 levels have been described numerous times (e.g., Lu et al., 2007; Frierson et al., 2007), but the implication of these mechanisms in the South American seasonality lowering zone appears non-obvious. Deeper analyses would be needed to understand the atmospheric dynamics in the simulations, which is out of the scope of the study"

We also modified the sentence l. 263 following your second suggestion:
"This phenomenon is well explained by two distinct chain reactions" by "This phenomenon could be explained by several chains of reaction, which are driven by both atmospheric and/or oceanic responses depending on the area".

[Figure]

**Additional diagnostics** - Annual variability of multiple climate parameters within the different seasonality lowering terrestrial zones between 3X and 2X (a-c,g,h): surface atmospheric temperature (black), latent heat flux (soil to atmosphere; brown), hydrological cycle (incl. precipitation, evaporation and net precipitation, different shades of blue), and net primary production (green). (d-f) Temperature changes and ΔMATR between the simulations. Rectangles contour terrestrial zones (ocean zones are not included) analysed in subfigures (a-c,g,h).

**2 Especially the argument of increasing cloud cover is not very convincing to me. In western Europe, there is a 10-20% increase, but that is not really seen in the southern hemisphere (small patches of 10% increase in austral summer). However, Fig 4b and magenta contours in Fig 5 seem to suggest that the negative MATR changes take place at the edge of the Hadley cell (and the associated ocean gyres/fronts) and the changes would be consistent with an equatorward/poleward shift of the Hadley cell – which would also impact the oceanic subpolar gyres. The Hadley cell extent has been well studied and can be related to changes in a latitudinal temperature gradient, which is clearly changing in these simulations. I would encourage the authors to rethink their results in this context.**

We have performed additional diagnostics but the response does not seem to explain the subtropical trends, and in particular: the summer cooling signal observed in South America (and the associated decrease in temperature seasonality).

We observe changes, particularly in austral summer (JFM), with an increase in the intensity of the Hadley cell and a slight southward shift in the rising limb of the cell (new Fig. S4, h,i), in agreement

with studies of the change in cell intensity and width under higher *p*CO2 (Lu et al., 2007; Frierson et al., 2007, and Chemke and Polvani, 2020, all three in *Geophysical Research Letters*). In parallel, there is a northward migration of the polar front (boundary between atmospheric polar cells and Ferrel cells), especially during the austral summer, and of the westerly wind maximum (by about 2° latitude, annually but less markedly during the austral winter, JAS; Figure S4). The Antarctic Circumpolar Current follows this northward shift (Figure S5), limiting the arrival of warm subtropical waters to the South Atlantic, between 40-45°S, but independently of the time of year.

The implication of changes in atmospheric and oceanic dynamics on temperature variability remains unclear as they have a small amplitude compared to the mid/high latitude anomalies and the latter, although small, would require a more detailed study which is - as you mention - out of scope here.

We may evoke these mechanisms in the results, in the form of two additional figures (Fig. S4 and S5 below), but nuancing their potential impact.

[Figure]

*New Figure S4.* Changes in atmospheric temperature and vertical circulation patterns between 3X and 2X in the Southern Hemisphere. (a,b) latitudinal surface temperature gradient; (c) zonal winds; (d-g) Air temperature (shaded), atmospheric cell extent (zonal mean streamfunction, lines) and vertical winds (arrows) in austral summer and winter for the simulations 3X and 2X. (h,i) Temperature, atmospheric cell extent and wind changes between the simulations 3X and 2X. The white arrow shows the northward migration of the polar/ferrel cell boundary. Dashed lines indicate anticlockwise circulation, solid lines, clockwise circulation. Arrows correspond to vertical winds. Atmospheric circulation was calculated over the pacific sector, between 180-30 °W.

[Figure]

(a) 3X  (b) 2X  (c) 3X - 2X

Water transport (m.s$^{-1}$)

0   0.04   0.08   0.12   0.16   0.20   0.24

Water transport changes (m.s$^{-1}$)

-0.06   -0.04   -0.02   0   0.02   0.04   0.06

*New Figure S5.* Annually 0–300 m depth averaged current velocity through the Southern Ocean (annual average, m.s$^{-1}$).

**3 In relation to comments #1-#2 I would encourage the authors to check the relative change in MATR. Since the MATR is usually small over the ocean, I would think that some of the signals would be emphasized, and maybe easier to appreciate, if one would look at the change relative to the baseline (i.e change in percentage).**

Thanks for this interesting comment. We completed Figure 4 with the relative changes in MATR for 2X-3X and 2X-ICE-SL - 3X, and modified Figure S6 to enable a direct quantification of relative MATR changes associated to each forcing. Results are consistent with our previous figures although high-latitude seasonality increases tend to look more moderate.

We also provide relative ΔMATR values in the text, sections 3.1.2, 3.1.3 and 3.1.4

- "It [The large MATR increase at high northern latitudes] represents an increase in MATR of 5-20% between 3X and 2X and up to 40% between 4X and 2X (Figure 4 e and S6 b)."

- "The widest zones with decreasing MATR are located within the 30-50°N latitudinal band, across North America, Western Europe, Central Asia, and 30-50°S for South America and Australia (depending on the $pCO_2$ lowering considered, 280 or 560 ppm, regionally up to 20 or 30% reduction of the MATR, Figure 4)."

- "As visible from relative ΔMATR, seasonality strengthening takes place both in areas characterized by strong or weak seasonality during the Eocene (Figure 4 f and S6 f)."

[Figure]

*Revised* **Figure 4.** Subfigures e,f now indicate relative ΔMATR, for 2X-3X and 2X-ICE-SL - 3X respectively (%).

[Figure]

*Revised* **Figure S6.** Additional ΔMATR maps. Left side maps (a,c,e) show absolute ΔMATR, while right side maps (b,d,f), relative changes (%).

**4 To me the proxy-data comparison mainly demonstrates that the simulations and proxies do not match in several locations in the 35-60N latitude band. I agree with the authors that especially in Europe the changing sea-level in the complex topography might be important (changing from sea to land would increase seasonality), and I wonder if it would be possible to 1) indicate which locations are in Europe in Fig. 7 and/or 2) provide a figure like S2 showing also the MATR difference in the proxy locations (coloring the dots accordingly).**

Thank you for this comment, for ease of reading, we propose to modify figures 7, S2 and Table S1 as follows. For Figure 7 (now Figure 8), we simplified the figure by suppressing subfigures b and c, to only keep the most realistic Priabonian-Rupelian scenario (2X-ICE-SL - 3X) and agree to indicate the id-number of the different localities (from subfigure a) to more easily identify the European sites. For Figure S2, we have added the Eocene MATR on subfigures (a,b), we also added two new columns to table S1 in order to provide initial MAT and MATR for the different localities.

[Figure]

*Revised* **Figure S2** with Eocene MATR values on subfigures a,b.

Minor:

**5 I think it would be easier to see that the MATR change is due to cool summers if the authors would show [2X-3X (JFM)]-[2X-3X (annual)] in the second row, and [2X-3X (JAS)]-[2X-3X (annual)] in the third row. At the moment one needs to do this comparison by eye, which is not optimal.**

Thanks for this comment which improves the visualization of our results, in the new figure dedicated to the areas of decreasing temperature seasonality (see above the new Fig. 6), we include three sub-figures which show the temperature change in JFM, in JAS, then the summer-winter temperature differences (JAS-JFM for the north, JFM-JAS for the south). This summer cooling is all the more visible.

**6 L424, L488: The authors write "The best representation of the temperature seasonality evolution from Priabonian to Rupelian arises when sea level drop is taken into account..." and "Europe stands in an intermediate position between North America and Asia with generally weaker changes in MATR.". It is unclear if these statements are based on the model results presented in this study (if yes, then please refer to figure/section in the manuscript) or is there some proxy/literature support as well (if yes, please provide references here).**

Thank you for noticing. These are based on model results; associated figures and tables are now given.

-   l. 427: "The best representation of the temperature seasonality evolution from Priabonian to Rupelian arises when sea level drop is taken into account (Table 1, Figure 8)"
-   l. 488: "Europe stands in an intermediate position between North America and Asia with generally weaker changes in MATR (Figure 4.d)".

Language/Typos:

**7 L135 'the' instead of 'a'**

Is it about "a narrow Southern Ocean gateways" ? We suppressed "a" to be more consistent with other geographic characteristics given earlier in the sentence.

Figures:

**8 Fig. 1: The authors might want to check how they save the image. In the pdf version, it seems that there are some longitudinal stripes that I believe are not realistic. This is not a huge issue, but it could be due to an artifact of switching between ps/pdf or something similar, so maybe worth checking if it can be easily fixed.**

Thank you. All the figures will be saved in .tiff which will ensure a good quality and prevent "stripe problems".

**9 Fig. 2: I would suggest adding 3X shoreline contour to panels using 2X-ICE_SL (d,h,i). I was a bit confused first about the large positive temperature differences, but then realized that those are in regions where the land-sea distribution has changed.**

This is a good idea, thank you, we modified subfigures 2d,h,i accordingly. Initial (i.e., before sea-level lowering) shorelines are now visible in magenta.

[Figure]

*Revised* **Figure 2.** After modification of subfigures d,h,l. Shorelines are visible in magenta.

**10 Fig 5. in panels e-f most of the latent heat flux change is negative, but in the text, the authors talk about an increase. I understand that this apparent contradiction can be simply due to a sign convention (negative down), but I would suggest flipping the sign (so positive anomaly implies an increase), and also define the sign of the fluxes in the caption. The same is true for other figures as well, I would ask the authors to use positive for an increase and negative for a decrease.**

There might be a misunderstanding here. Over the ocean, the majority of the signal is negative but continental areas mostly exhibit an increase in latent heat fluxes. In fact, the increase in precipitation in the 50°N-50°S area causes an increase in latent heat fluxes and primary productivity in summer (see especially the area north of Australia in sub-figures 5.e,g and North America and Asia 5.f,h).

For more clarity, this figure was replaced by a new figure, now Figure 6. (see answer to comment #1)

**11 Fig 6: L295, I believe the authors mean 'low-level cloud fraction changes'**

Indeed, thanks for noticing. This was corrected.

**DATA ADDITION**

In correcting our manuscript, we felt it was unfortunate not to include data from the compilation of Pound and Salzmann 2017, given the small number of proxy-data available. We propose to increase the number of data and thus change the results accordingly as presented hereafter. We selected data from Pound and Salzmann, 2017 to retain (1) the best dated data according to the dating quality indicator used by their study (data Q1 to Q3), (2) sites with temperature estimates for the Priabonian and Rupelian, or at least one nearby locality that could be compared. No Eocene-Oligocene site was selected for more clarity. This allowed us to add 18 data points (to the 17 points present in v1 of our publication). In an effort to limit the addition of overly uncertain ΔMATR data, we chose not to include data with a range of CMMT estimates (CMMTmax - CMMTmin) ≥ 10°C (either for Priabonian or Rupelian sites). Of these new sites, 14 are located on the continents and enable a direct comparison to model ΔMATR values, the others from marine cores using pollen of uncertain provenance, are shown in the new Figure 4 but are not used in the statistical analyses.

For greater realism, we also changed the way we calculated the differences in ΔMATRmin and ΔMATRmax (i.e., the negative and positive error associated to ΔMATR from the data), which did not sufficiently reflect the possible extent of ΔMATR. ΔMATRmin/max are now calculated from the average prediction error of the coldest (CMMT) and warmest (WMMT) months, instead of simply the difference between ΔMATRmin and ΔMATRmax (see below).

In the submitted version of the manuscript
ΔMATRmin = MATRmin(recent) – MATRmin(old)
ΔMATRmax = MATRmax(recent) – MATRmax(old)

In the new version
Error ΔMATR = average((CMMTmax-CMMTmin)+(WMMTmax-WMMTmin))

**RMSE analysis -** The addition of these data decreases the average model-data difference and leads to better RMSE scores as well (see Table 2). It is nevertheless necessary to specify that, for the RMSE, this low deviation is partly due to the sometimes-wide prediction ranges of ΔMATR (difference between ΔMATRmin and ΔMATRmax). The trends described in the first version of the paper remain the same with a slightly reduced prediction when the Antarctic ice-sheet alone is added, but the best-one when the Antarctic ice-sheet and sea level decrease are added together.
In addition, a better agreement between data and simulations without sea level drop is also observed, as visible with the percentage of sites where the direction of ΔMATR is adequately modelled (Table 2 below, line "%"). This is due to data points from Pound and Salzmann (2017) predicting decreases in ΔMATR in areas where the model also predicts a decrease in seasonality (which is based, as explained in v1 of the manuscript, on the lowering of pCO2). As before, agreement is better when the least warm Eocene simulation (3X) is used as the reference point for the model's ΔMATR calculation (right part of Table 2).

Table 2 – Grey values are from the original manuscript, blue values are new values calculated after adding new data from Pound and Salzmann, 2017.

| | 2X - 4X | 2X-ICE - 4X | 2X-ICE-SL -4X | 2X - 3X | 2X-ICE - 3X | 2X-ICE-SL -3X |
|---|---|---|---|---|---|---|
| Mean ΔMATR (model - data) | 5.3 °C | 5.8°C | 3.9°C | 4.6°C | 5.1°C | 3.2°C |
| | **-3,52°C** | **-3,91°C** | **-1,92°C** | **-2,81°C** | **-3,20°C** | **-1,20°C** |
| RMSE | 5.0°C | 5.3°C | 4.1°C | 4.8°C | 5.0°C | 3.8°C |
| | **3,06°C** | **3,38°C** | **2,49°C** | **2,91°C** | **3,15°C** | **2,35°C** |
| % | 5,8 % | 5,8 % | 35,3 % | 0,0 % | 0,0 % | 58,8 % |
| | **19,35%** | **19,35%** | **41,94%** | **22,58%** | **16,13%** | **45,16%** |
| rho | 0.21 (p = 0.45) | 0.35 (p = 0.20) | 0.57** (p = 0.02) | 0.20 (p = 0.47) | 0.37 (p = 0.17) | 0.56** (p = 0.03) |
| | **0.21 (p =0.28)** | **0.27 (p = 0.16)** | **0.29 (p = 0.12)** | **0.19 (p = 0.32)** | **0.25 (p = 0.20)** | **0.29 (p = 0.12)** |

Note: In the submitted version of the paper, the line "mean MATR" was providing absolute changes between model and data, we now show the sign of the difference to be more informative (i.e. to show that the model slightly underpredict ΔMATR changes).

**Correlation** – Adding Pounds and Salzmann (2017) points, removes the correlation of the ΔMATRs of the model and the data (even with a Pearson parametric correlation test). While a lack of correlation is always a bit disappointing, we do not believe that it discredits our approach of adding more data. It is certain that a study with more data will be more reliable. Although there is no statistical correlation, the data visualized on the map (Fig. 5) and the RMSEs show a rather encouraging agreement, and it is not surprising that mismatches may exist due to errors in the data, paleolatitude reconstruction, temperature gradient modeling that may influence the agreement between ΔMATR of the model and data.

This lack of correlation seems to be largely explained by only 5 points (see figure below), without which, a significant model data correlation is restored (rho = 0.54, p-value = 0.007). It is not within our competence nor within the scope of the paper to re-analyze these data. It could also be that some of the proxy datasets ("never matching points") point to major inherent biases in fossil plant assemblages (sampling bias, taphonomic bias, methodological bias of paleoclimate estimation...), while the paleoclimate estimations are accurately done. The issue of such discrepancies can't be resolved until plant-independent paleoclimate proxy data is available for such sites to confirm or not plant-based paleoclimate estimations. Two of these points are in Europe (in addition to the two qualitative points not included in the statistical analyses) and question our ability to reconstruct the seasonality of this fragmented continental area with the spatial resolution of the model.

Finally, reanalysis of the MATR change data allowed us to show that 90-100% of the data describing no MATR changes are located in areas where the model predicts a decrease in ΔMATR following the decrease in pCO2 (comparison of 2X-4X, and 2X-3X simulations, respectively), which seems to

support our hypothesis made in v1 of the paper that changes in the taxonomic composition of vegetation may not necessarily reflect decreasing seasonality of temperatures.

[Figure]

New Priabonian-Rupelian ΔMATR compilation

---

## Author Response (AR1)

Response to reviewers

Manuscript CP-2021-27

**Evolution of continental temperature seasonality from the Eocene greenhouse to the Oligocene icehouse - A model-data comparison**

Agathe Toumoulin, Delphine Tardif, Yannick Donnadieu, Alexis Licht, Jean-Baptiste Ladant, Lutz Kunzmann, and Guillaume Dupont-Nivet
* * *
Rouen, November 10[th], 2021

Dear Dr. Alberto Reyes and reviewers,

We sincerely thank you and the reviewers for feedback and the opportunity to present the attached revised version of our manuscript. We have carefully accounted for all the reviewer comments in the revised manuscript, following the reply to reviewers we posted on July 27[th], (https://cp.copernicus.org/preprints/cp-2021-27/#discussion) which explains both documents have important similarities. In the present document, reviewer's comments are black written in black and our answers, in blue. The line numbers given are those of the tracked version of the manuscript.

Important remarks by RC1, concerned the possible mechanisms associated with areas of decreased temperature seasonality and the representation of these results. To address this, we made new diagnostics which we included as figures and supporting figures in the manuscript.

As explained during the discussion process, while correcting our manuscript, we felt it was unfortunate not to include data from the compilation of Pound and Salzmann (2017), given the small number of data available. Thus, we increased the number of data in the paper (14 more data-comparison points, see updated Figure 4 and 8) and updated the manuscript accordingly following the interactive discussion. We are aware that this kind of practice is not usual, and we apologize for the extra work it may require, but we believe it will give more representative picture of seasonality changes through the Eocene-Oligocene Transition. The message of the paper and the conclusions remain the same.

Kind regards,

Agathe Toumoulin, on behalf of all co-authors
* * *
Response to Reviewer 1

**Major comments:**

**1 Overall, I think 3.1.3 is cutting corners, it might well be that some of the stated mechanisms are true, but it is difficult (if not impossible) to confirm the mechanisms based on the evidence presented. For this type of paper, it is not crucial to identify the exact mechanism, although it is valuable of course. I would suggest the following 1) to be able to make a bit more robust statement, the authors could check the correlation between surface air temperature change and latent heat flux change/P-E change/Primary prod. Change. 2) I would change the language towards 'we suggest that this phenomenon could be explained by...' rather than 'this phenomenon is well explained by'.**

Thank you for this comment. We have performed additional diagnostics to better evaluate the potential mechanisms at stakes in our simulations. We focused particularly on the regions demonstrating a decreased seasonality in the early Oligocene (mostly at mid-latitudes), given that they constitute the most counter-intuitive result of this study. To do so, we extracted the anomalies in Precipitations, Evaporation, resulting P-E, NPP and surface temperature on land areas between 2X and 3X simulations (since the areas of decreasing seasonality appear as a result of decreasing $CO_2$).

We feel no evident correlation or single mechanism emerges from these diagnostics and the magnitude of these changes is highly variable from one region to the other which supports that decreasing

temperature seasonality may result from various mechanisms depending on the considered area. The agreement between decreasing summer temperatures and increasing latent heat fluxes/net precipitation appears particularly good over the United States, and less so over Asia and Australia. In contrast, additional mechanisms are needed to explain the temperature changes over Europe and southern South America (as mentioned in the first version of the manuscript, see also our answer to your next comment). Finer analysis involving perhaps daily to hourly resolution might be necessary to provide a better understanding of the mechanisms at stake

- We have restricted Fig. 5 to subfigures (a-d) and made a new figure (new Figure 6) showing co-variations between temperature, latent heat, hydrological cycle (precipitation / net precipitation / evaporation), and net primary productivity. We included it in the manuscript (instead of as supporting material).

- We reformulated the sentence originally located l. 263 (l. 297 in tracked MS):

(Original sentence) *"Temperature changes are characterized by polar amplification, with a stronger winter cooling at high-latitudes (Figure 2. a,b,e,f,i,j). This phenomenon is well explained by the combined effect of albedo and sea-ice feedback."*

(New sentence, l. 297) *"Temperature changes are characterized by polar amplification, with a stronger winter cooling at high-latitudes (Figure 2. a,b,e,f,i,j), likely due to the combined effect of albedo and sea-ice feedback."*

**2 Especially the argument of increasing cloud cover is not very convincing to me. In western Europe, there is a 10-20% increase, but that is not really seen in the southern hemisphere (small patches of 10% increase in austral summer). However, Fig 4b and magenta contours in Fig 5 seem to suggest that the negative MATR changes take place at the edge of the **Hadley cell (and the associated ocean gyres/fronts)** and the changes would be consistent with an equatorward/poleward shift of the Hadley cell – which would also impact the oceanic subpolar gyres. The Hadley cell extent has been well studied and can be related to changes in a latitudinal temperature gradient, which is clearly changing in these simulations. I would encourage the authors to rethink their results in this context.**

We have performed additional diagnostics but the response does not seem to explain the subtropical trends, and in particular: the summer cooling signal observed in South America (and the associated decrease in temperature seasonality).

We observe changes, particularly in austral summer (JFM), with an increase in the intensity of the Hadley cell and a slight southward shift in the rising limb of the cell (new Fig. S4, h,i), in agreement with studies of the change in cell intensity and width under higher pCO2 (Lu et al., 2007; Frierson et al., 2007, and Chemke and Polvani, 2020, all three in *Geophysical Research Letters*). In parallel, there is a northward migration of the polar front (boundary between atmospheric polar cells and Ferrel cells), especially during the austral summer, and of the westerly wind maximum (by about 2° latitude, annually but less markedly during the austral winter, JAS; Figure S4). The Antarctic Circumpolar Current follows this northward shift (Figure S5), limiting the arrival of warm subtropical waters to the South Atlantic, between 40-45°S, but independently of the time of year. The implication of changes in atmospheric and oceanic dynamics on temperature variability remains unclear as they have a small amplitude compared to the mid/high latitude anomalies and the latter, although small, would require a more detailed study which is - as you mention - out of scope here.

- We made two additional supporting figures (Fig. S7 and S8 below), and now evoke these mechanisms in the result and discussion (l. 522-527) but nuancing their potential impact.

(Results, l. 354-368) *For southern South America, several parameters seem consistent with the reduction of the MATR but it is difficult to disentangle their contribution. By amplifying the latitudinal temperature gradients, the $pCO_2$ drop induces a northward migration of the westerly wind maximum (by about 2° of latitude, annually but less markedly during austral winter, JAS) and of the Antarctic*

*Circumpolar current, which delimits the southern hemisphere subpolar and subtropical gyres. This northward shift thereby limits the arrival of warm subtropical waters towards the poles (Figure S7). This greater cooling in summer SST reinforces the ocean's buffering effect on atmospheric temperatures in southern South America and favors milder summers, and to a lesser extent, cooler winters, which is consistent with a decrease in seasonality (Figure 7). Finally, changes in atmospheric dynamics (decrease in the width and increase in the intensity of the Hadley cell) are also visible and could have an impact on air-ocean exchanges, but much more analysis would be needed to understand their implication, which is not the focus of this paper (Fig. S5).*

*(Discussion, l. 522-527) In parallel, although the implication of changes in the atmospheric circulation in the southern South American seasonality lowering zone appears non-obvious, the intensification and weakening of the Hadley cell extent in relation to changing $pCO_2$ levels have been described numerous times (e.g., Lu et al., 2007; Frierson et al., 2007). Deeper analyses would be needed to understand the atmospheric dynamics in the simulations, which is out of the scope of the study*

**3 In relation to comments #1-#2 I would encourage the authors to check the relative change in MATR. Since the MATR is usually small over the ocean, I would think that some of the signals would be emphasized, and maybe easier to appreciate, if one would look at the change relative to the baseline (i.e change in percentage).**

Thanks for this interesting comment.

- We completed Figure 4 with the relative changes in MATR for 2X-3X and 2X-ICE-SL - 3X,
- and modified Figure S6 to enable a direct quantification of relative MATR changes associated to each forcing. Results are consistent with our previous figures although high-latitude seasonality increases tend to look more moderate.
- We also provide relative ΔMATR values in the text (e.g., l. 303-304 and 330-332)

**4 To me the proxy-data comparison mainly demonstrates that the simulations and proxies do not match in several locations in the 35-60N latitude band. I agree with the authors that especially in Europe the changing sea-level in the complex topography might be important (changing from sea to land would increase seasonality), and I wonder if it would be possible to 1) indicate which locations are in Europe in Fig. 7 and/or 2) provide a figure like S2 showing also the MATR difference in the proxy locations (coloring the dots accordingly).**

Thank you for this comment, for ease of reading, we propose to modify figures 7, S2 and Table S1 as follows.

- We simplified the figure 7 (now Figure 8) by suppressing subfigures b and c, to only keep the most realistic Priabonian-Rupelian scenario (2X-ICE-SL - 3X). All points have been given a number so they can be associated to the different localities shown in subfigure a. It will easily permit the identification of European sites.
- A similar figure is available in supporting material (Figure S9), which shows ΔMATR with the simulation 4X for the Eocene.
- We completed Figure S2 (subfigures a,b) and Table S1, with Eocene in order to provide initial MATR (and for Table S1 MAT) for the different localities.

**Minor:**

**5 I think it would be easier to see that the MATR change is due to cool summers if the authors would show [2X-3X (JFM)]-[2X-3X (annual)] in the second row, and [2X-3X (JAS)]-[2X-3X (annual)] in the third row. At the moment one needs to do this comparison by eye, which is not optimal.**

Thank you for this comment which improves the visualization of our results.

- In the new Figure 6, we include three subfigures (d-f) which show the temperature change in JFM, in JAS, then the summer-winter temperature differences (JAS-JFM for the north, JFM-JAS for the south). This summer cooling is now more visible (subfigure f).

**6 L424, L488: The authors write "The best representation of the temperature seasonality evolution from Priabonian to Rupelian arises when sea level drop is taken into account..." and "Europe stands in an intermediate position between North America and Asia with generally weaker changes in MATR.". It is unclear if these statements are based on the model results presented in this study (if yes, then please refer to figure/section in the manuscript) or is there some proxy/literature support as well (if yes, please provide references here).**

Thank you for noticing. These are based on model results; associated figures and tables are now given.

- l. 568: "The evolution of temperature seasonality from the Priabonian to the Rupelian is better represented when the sea level drop associated with the AIS is taken into account (Table 2, Figures 4, 8)."
- l. 488: "Europe stands in an intermediate position between North America and Asia with generally weaker changes in MATR (Figure 4 d);".

**Language/Typos:**

**7 L135 'the' instead of 'a'**

Is it about "a narrow Southern Ocean gateways"? We suppressed "a" to be more consistent with other geographic characteristics given earlier in the sentence.

**Figures:**

**8 Fig. 1: The authors might want to check how they save the image. In the pdf version, it seems that there are some longitudinal stripes that I believe are not realistic. This is not a huge issue, but it could be due to an artifact of switching between ps/pdf or something similar, so maybe worth checking if it can be easily fixed.**

Thank you.

- All the figures will be saved in .tiff which will ensure a good quality and prevent "stripe problems".

**9 Fig. 2: I would suggest adding 3X shoreline contour to panels using 2X-ICE_SL (d,h,i). I was a bit confused first about the large positive temperature differences, but then realized that those are in regions where the land-sea distribution has changed.**

This is a good idea, thank you.

- we modified subfigures 2d,h,i accordingly. Initial (i.e., before sea-level lowering) shorelines are now visible in magenta.

**10 Fig 5. in panels e-f most of the latent heat flux change is negative, but in the text, the authors talk about an increase. I understand that this apparent contradiction can be simply due to a sign convention (negative down), but I would suggest flipping the sign (so positive anomaly implies an increase), and also define the sign of the fluxes in the caption. The same is true for other figures as well, I would ask the authors to use positive for an increase and negative for a decrease.**

There might be a misunderstanding here. Over the ocean, the majority of the signal is negative but continental areas mostly exhibit an increase in latent heat fluxes. In fact, the increase in precipitation in the 50°N-50°S area causes an increase in latent heat fluxes and primary productivity in summer. This will be clearer with the new Figure 6 (see also answer to comment #1).

**11 Fig 6: L295, I believe the authors mean 'low-level cloud fraction changes'**

Indeed, thanks for noticing. This was corrected.
* * *
Response to Reviewer 2

**1 I was not wholly satisfied with the introduction. The themes and content of the introductory sections are generally appropriate, however, I feel that their organization and connectivity could be improved. For example, I felt the context of the EOT as provided in section 1.1 was a bit shallow. The chance to set the stage of the EOT is somewhat lost as the authors transition very quickly into how temperature seasonality can be quantified. I think there is an opportunity to offer more to the reader about our current understanding of the EOT and the significance of the event as a potential analogue with respect to our modern climate. Some of these ideas are presented at the end in the conclusions, but I think they could be presented earlier.**

The aims of the study are provided in section 1.4; however, the overall placement of this section feels late. I was left wondering very early as I was reading through sections 1.1 through 1.3 what the authors were planning to accomplish. I think presenting this earlier will provide better context to the reader for what the authors goals are as they read through the following sections. I would suggest the authors to consider revising the introduction to improve some of these shortcomings.

Thank you for this comment. We agree the paleoclimatic context was a bit too short. We went back to the introduction to better contextualize our study and now announce research questions earlier in the text, as you suggested.

(l. 41-46) *"The Eocene-Oligocene Transition (EOT) is marked by an abrupt cooling event (~2.9°C from marine proxies; Hutchinson et al., 2021), considered as the hinge between the Eocene greenhouse and the later Cenozoic icehouse. This event is associated with the first major expansion of the Antarctic ice-sheet with an estimated sea level drop of ~70 m (Hutchinson et al., 2021; Miller et al., 2020). The EOT is described as a relatively brief event (~790 000 years), with two successive steps (at ca. 33.9 and 33.7) recognized in extensively studied marine environments, especially from deep ocean $\delta^{18}O$ values (e.g. Katz et al., 2008; Zachos et al., 2001; see the review of Hutchinson et al., 2021)."*

New text at the end of section 1.1 (l. 67-71) *"By comparing paleoclimate simulations to a synthesis of indicators of seasonality changes (Table S1), our study attempts to reconstruct the evolution of seasonal temperature contrast from the middle Eocene to the early Oligocene. The EOT is broken down into five simulations, describing the evolution of three major forcing at that time: the $pCO_2$ drawdown, the AIS expansion and the resulting sea-level lowering, in order to understand the respective contribution of each component on the resulting seasonality change patterns, along with their possible synergies and retroactions."*

Regarding the association of our results with the current climate deterioration context, we kindly disagree and would prefer not to do so. You are right, the study of ancient warm climates is often justified by the current climate crisis, the Miocene Climatic Optimum has also been called a potential model for our future world. Yet, although $pCO_2$ already reached values reconstructed for the Oligocene we consider highly speculative that we may face a change back to the Eocene world if we reach late

Eocene pCO2 values, notably because of different geographies. We thus would prefer to conservatively restrict the scope of our study modeling the Eocene climate to the mechanisms of greenhouse climate in line with previous studies on that period.

**2 In section 1.2 the authors list a number of plant genera and family, however, only in a couple cases are a more common or generalized named provided. Not all readers may be familiar with the plant genera or families listed and thus some quickly communicated information about the type of habitats that these plants represent is lost. This becomes especially problematic when plant families that are no longer formally recognized, such as Flacourtiaceae, are used. This makes it especially difficult if a reader tries to discover more. I would recommend the authors provide the common names for the listed genera and families as this can only help the botanically unfamiliar reader.**

Thank you for this comment. We agree on providing common names to simplify. For the beginning of the sentence, we modified the order of the words within the sentence to provide common names first, following your view. However, we kept some of them in the end of the sentence because there is not always a common name for families and genera. Apart from laurel, common names may not be particularly enlightening to non-botanist reader anyway (e.g. one could talk about "annonaceae" by saying "annonas" and "myrtaceae" by saying "myrtles"). Also, since Flacourtiaceae was divided into various different families, we deleted it.

*Original text (l. 64-69) "[...] species characteristic of warm paratropical to temperate environments such as conifers Doliostrobus sp. (conifers), Nypa sp. (palms), Rhodomyrtophyllum sp. (Myrtaceae), and some families with tropical elements such as Annonaceae, Lauraceae, Cornaceae, Flacourtiaceae, Icacinaceae, Menispermaceae, and, depending on bioclimatic zones, the expansion of temperate to boreal vegetation through the increase of deciduous and / or coniferous species (Eldrett et al., 2009; Kunzmann et al., 2016; Kvaček, 2010; Kvaček et al., 2014; Mosbrugger et al., 2005; Utescher et al., 2015; Wolfe, 1992)."*

*New text (lines 87-93)): "[...] species characteristic of warm paratropical to temperate environments such as palms (e.g., Nypa sp.), plants from myrtle and eucalyptus family (Myrtaceae, e.g., Rhodomyrtophyllum sp.), conifers (e.g., Doliostrobus sp.) and some plant families with tropical elements (e.g., Annonaceae, Lauraceae, Cornaceae, Icacinaceae, Menispermaceae), and, depending on the bioclimatic zones, the expansion of temperate to boreal vegetation through the increase of deciduous and / or coniferous species (Eldrett et al., 2009; Kunzmann et al., 2016; Kvaček, 2010; Kvaček et al., 2014; Mosbrugger et al., 2005; Utescher et al., 2015; Wolfe, 1992)."*

**3 In figure 5 panels g-h the model simulations show changes in primary productivity. These panels as ordered imply to me that the model is suggesting that primary productivity increased in the northing latitudes during the summer (JAS). I am not sure if there is a convention here that is being used that I am unfamiliar with, but if this is not the case and model does show a decrease in net primary productivity then this would be very counter-intuitive to what is expected and requires some explanation. This also seems contradictory to what is stated in the text in section 3.1.3, where the authors state that conditions favour primary productivity in the summer.**

Thank you for your comment, our figure was indeed confusing. In fact, we are talking about the increase in primary productivity within the areas of decrease in MATR, which are framed by the pink dotted lines in the original Figure 5.

We modified this figure as follow (see also our response to comment#1 of the other reviewer):

- Restricting Fig. 5 to subfigures (a-d)
- Making a new figure (Figure 6) showing temperature, latent heat, hydrological cycle (precipitation / net precipitation / evaporation), and net primary productivity changes between 2X and 3X, with regional plots (one for each zone in which MATR decreases) instead of maps

(as in the original Figure 5). It is now easier to identify eventual correlations between the different parameters.

**4 In table 1 the authors defined MAT as the Mean Annual global 2-meter air Temperature, which appears to add an additional layer of complexity to the well-known definition of MAT. Although this is a relatively minor point, I would suggest better to call it Global MAT or devise a different acronym for this purpose. This usage is also different to how MAT is defined by the authors in supporting table S1. For Table S1 MAT is defined as the average Mean Annual Temperature changes. I think it would be better for this table S1 to be labeled as ΔMAT. There needs to be consistency between definition used in both the manuscript and the supplemental information.**

Thank you for noticing. It is important to keep the information available that model MAT values were obtained from a variable calculating the temperature at a 2m height (for precision and result's replicability purpose).

- We took care being more precise when providing temperatures throughout the manuscript so there is no ambiguity.
- We used ΔMAT for Table S1. In the same way, we added a "Δ" to the headings of the other columns of the table S1.

**5 There is not much discussion about the paleogeographic position of the proxy data used to compare against the model simulations. The locations of the fossil proxy localities are important to the context of the changing sea level. If the forests that the plants were growing in were affected by a coastal climate, then a reduction in sea level would have greatly influenced seasonality and promoted a more continental climate. However, if some of these localities were already far away from a coastline, they may not have experienced a significant increase in seasonality. Coastal influence is discussed briefly, but a greater context I feel is absent and think would add to the authors discussion.**

The effect of transgressions/regressions on the change in oceanity/continentality of regional climate and thus regional vegetation is an interesting point.

In the original version of the MS, we suggested this effect in the section 4.1.2: *"Interestingly, the combination of the three forcing mechanisms also lead to a better agreement of modelled ΔMATR and middle to late Eocene data, especially in coastal areas of Kamchatka, and South China (triangles, Figure 4). Although the 70-m sea level decrease from the 2X-ICE-SL simulation is too important for the late Eocene, the better data-model agreement when both AIS and sea-level decrease are considered suggests that small ice-sheet development before the EOT may have played a significant role in driving the middle to late Eocene ΔMATR."*

Even if we agree with you on the principle that the vegetation of coastal areas has certainly been more affected by the drop-in sea level than continental areas, which we suggest in our article, it could also be that in some areas, even coastal ones, the drop-in sea level is not systematically recorded by the vegetation considering that we have only 10-20% of the original vegetation (woody species) preserved as fossils in macro-floras. However, and this is important, what paleobotanists have recognized is that floristic composition could indeed be markedly different between neighboring lowland coastal plain regions depending if they are influenced by different-warmed seas/oceans. So, we could hypothesize that it is particularly the disappearance of temperate to warm shallow basins (and epicontinental seas) that should be recorded more frequently. However, a much larger number of points would be needed to confirm this.

We completed our discussion with the following sentences:

- (l. 571-573): *"Our results are very dependent on the paleogeography used in the simulations and of the proxy location used in our data-model comparison."*

**Response to reviewers**

- (l. 581-587): *"The heterogeneity shown in data might thus result from smaller scale paleogeographic changes through the EOT that are not well represented by the resolution used in our simulations. This variability of data ΔMATR estimate could also be due to (1) a variable quality of MATR data related to the fragmentary nature of the fossil record and to differential recording of vegetation types, as well as (2) differences in the temperature of marine/oceanic zones before regression. Depending on their extension and depth, these seas may have buffered more or less importantly seasonal temperature variations of the nearby regions, and therefore their disappearance may have affected the MATR in different magnitude."*

---

## Author Response (AR2)

AUTHOR'S RESPONSE

**Manuscript CP-2021-27**

*Evolution of continental temperature seasonality from the Eocene greenhouse to the Oligocene icehouse - A model-data comparison*

Agathe Toumoulin, Delphine Tardif, Yannick Donnadieu, Alexis Licht, Jean-Baptiste Ladant, Lutz Kunzmann, and Guillaume Dupont-Nivet

Rouen, January 13th, 2022

Dear Dr. Alberto Reyes and reviewer,

We sincerely thank you for your feedback (and for having worked on the evaluation of our manuscript both during summer and Christmas vacations).

We have modified the abstract by changing the sentence:
" *Our simulations suggest that MATR changes across the EOT reflect the combined effects of pCO2 decrease, Antarctic glaciation and increased continentality, which result in zonal to regional climate responses.*"
by
" *Our simulations suggest that pCO2 lowering alone is not sufficient to explain the seasonality evolution described by the data through the EOT, but rather that the combined effects of pCO2, AIS formation, and increased continentality provide the best data-model agreement.*"

That's better, thank you for this suggestion.

Have a nice day and thank you again,

Bests,

Agathe, on behalf of all co-authors